# Fine-tuning language models to find agreement among humans with diverse preferences

**Michiel A. Bakker**[*]
DeepMind
miba@deepmind.com

**Martin J. Chadwick**[*]
DeepMind
martin@deepmind.com

**Hannah R. Sheahan**[*]
DeepMind
hsheahan@deepmind.com

**Michael Henry Tessler**
DeepMind
tesslerm@deepmind.com

**Lucy Campbell-Gillingham**
DeepMind
lcgillingham@deepmind.com

**Jan Balaguer**
DeepMind
jua@deepmind.com

**Nat McAleese**
DeepMind
nmca@deepmind.com

**Amelia Glaese**
DeepMind
glamia@deepmind.com

**John Aslanides**
DeepMind
jaslanides@deepmind.com

**Matthew M. Botvinick**
DeepMind
University College London
botvinick@deepmind.com

**Christopher Summerfield**
DeepMind
University of Oxford
csummerfield@deepmind.com

## Abstract

Recent work in large language modeling (LLMs) has used fine-tuning to align outputs with the preferences of a prototypical user. This work assumes that human preferences are static and homogeneous across individuals, so that aligning to a a single "generic" user will confer more general alignment. Here, we embrace the heterogeneity of human preferences to consider a different challenge: how might a machine help people with diverse views find agreement? We fine-tune a 70 billion parameter LLM to generate statements that maximize the expected approval for a group of people with potentially diverse opinions. Human participants provide written opinions on thousands of questions touching on moral and political issues (e.g., "should we raise taxes on the rich?"), and rate the LLM's generated candidate consensus statements for agreement and quality. A reward model is then trained to predict individual preferences, enabling it to quantify and rank consensus statements in terms of their appeal to the overall group, defined according to different aggregation (social welfare) functions. The model produces consensus statements that are preferred by human users over those from prompted LLMs ($> 70\%$) and significantly outperforms a tight fine-tuned baseline that lacks the final ranking step. Further, our best model's consensus statements are preferred over the best human-generated opinions ($> 65\%$). We find that when we silently constructed consensus statements from only a subset of group members, those who were excluded were more likely to dissent, revealing the sensitivity of the consensus to individual contributions. These results highlight the potential to use LLMs to help groups of humans align their values with one another.

---

[*]Authors contributed equally to this work

36th Conference on Neural Information Processing Systems (NeurIPS 2022).

# 1 Introduction

Modern large-scale transformer-based language models have revolutionized the capacity of AI systems to perform complex natural language processing tasks including reading comprehension, common sense reasoning, and fluent language generation [8, 27, 11, 17]. A key challenge in language modelling is to ensure that the generated text is helpful, legitimate, and aligned with human values [20, 34, 4, 14]. One popular approach is to recruit human participants to rate or compare candidate model outputs, providing feedback to the model about its performance on tasks like summarisation, instruction-following, and question answering [5, 26, 30, 37, 23, 15]. For large models, this approach improves performance on datasets specifically designed to test alignment (e.g., the HHH dataset), without decreasing their overall language competencies [4].

Though powerful and general, extant methods for fine-tuning language models from human preferences treat these preferences as if they were homogeneous and static. This assumption is reasonable for a task such as article summarisation, where there is an objectively defined ground truth (i.e., facts in the article that the model must summarise). However, for a wide variety of social problems that humans solve themselves using language (e.g., social coordination and group decision making), we cannot assume that people all share the same values. A key case study in alignment of diverse preferences is consensus formation. Consensus is commonly defined as the agreement of a large fraction of a social group about a particular topic or course of action. It is both a prerequisite for cooperation and a key pillar of the democratic process. Finding consensus for humans is not easy, and technology often exacerbates political division rather than fostering reconciliation among those with divergent opinions [3]. Large language models achieve surprising fluency and sensitivity to homogeneous preferences, but their ability to help people find agreement has not yet been tested.

Here, we investigate the use of large language models (LLMs) to aid humans in collectively producing written opinions that maximize approval rates among users. Specifically, we create a corpus of thousands of questions concerning political issues relevant in the United Kingdom, about which reasonable, well-informed people might legitimately disagree (for example, "Should we tax unhealthy foods and sugary drinks?", or "Should we re-nationalise the railways?"). We recruit groups of human participants to write out their opinions about these questions. We then fine-tune a 70 billion parameter language model (*Chinchilla* [17]) to produce candidate consensus statements that small groups of participants would be likely to endorse (see Table 1), guided by an underlying family of social welfare functions. Critically, in our work, the language model is not trained to adopt a particular opinion or persuade others of any one view. Rather, it is trained to produce consensus candidates based on the opinions contributed by the human group. We find that our particular data collection and training pipeline results in a model that generates statements that are preferred more strongly than a number of high-performing baselines, including individual human opinions. The statements generated by the model reflect the underlying opinions that are contributed by the users. This work opens up new possibilities for LLMs to help people find common ground in service of collective decision making.

# 2 Related Work

**Aligning LLMs with human preferences**. Several papers have reported the successful training of AI systems directly from human preferences (e.g., training deep RL agents to play Atari [12, 18]). In the domain of language modelling, human preferences have been used to fine-tune models to perform specific tasks, such as summarisation or translation [37, 30, 5], to provide inline citations supporting their claims [25, 23, 32], or for generic instruction following [26]. Our pipeline shares several design similarities with these works (e.g., the reward modelling approach). A key point of distinction, however, stems from the source of legitimacy in which statements produced by the language model are ostensibly grounded. For these related works, external sources that are generally agreed to be truthful and unbiased are the source of legitimacy. In our work, by contrast, we optimise, through a combination of opinion-conditional reward modelling and group-level welfare maximisation, to produce statements that will generate agreement for a specific group.

**Summarization**. Our work builds upon attempts to build natural language processing models to provide high-quality summaries of text (e.g., [10, 29]). These efforts have redoubled recently with the advent of LLMs [37, 30, 27]. Like consensus opinions, good summaries should be concise, informative and balanced passages of text. However, summaries and consensuses differ in their grounding: good summaries are based on accurate, fact-based information about the world (for which

| Question | Should we lower the speed limit on roads? |
|---|---|
| **Opinion 1** *participant* | The higher speed limits are ok and need no change. Some of the lower limits, like 30mph in built up areas, should be reduced and better enforced to reduce accidents. All in all the higher limits would not have much impact if reduced. |
| **Opinion 2** *participant* | Yes. Driving faster causes higher pollution and does not significantly shorten the average length of journeys. It would increase the fuel efficiency of those driving and prevent the distribution of microscopic particles of rubber from tyre wear into the environment. The number of road fatalities would reduce too. |
| **Opinion 3** *participant* | I think the speed limits are mainly fine the way they are. In our area speed limits are reduced around busy areas such as schools and parks which is a good thing. |
| **Consensus** *SFT-Utilitarian* | We believe that speed limits are currently appropriate. However, we feel that there is a need to enforce speed limits more strictly, particularly in areas with a high density of pedestrians such as near schools. We also feel that there is a need to educate people more about the effects of driving too fast, such as the effects on fuel efficiency and pollution. |

Table 1: A candidate consensus statement generated by our SFT-Utilitarian model in response to a question and corresponding opinions from the test set. The example was chosen randomly from a set of consensus statements that included 3 opinions and that was in response to a divisive question (i.e., questions for which there was some disagreement). Other examples can be found in the Tables A6-7.

a single viewpoint is typically accepted) whereas a consensus is grounded in the opinions of the specific individuals in a group seeking to achieve agreement. Our work specifically builds on work in *opinion summarisation*, in which subjective opinions (typically reviews about products, restaurants, or movies) are summarized into a kind of meta-review (e.g., [1, 31]). While similar in spirit to our work on consensus generation, extant work in opinion summarisation falls short of using actual human feedback and does not verify generated summarisations with the same individuals who provided their opinions, instead focusing on *third-party* evaluations (e.g., does this summary make sense in light of these opinions?). By not engaging with the people that provided the original opinions, these projects do not pertain to alignment problems but instead are closer to other summarisation work.

**Collective Reasoning**. Our work is relevant to a fast-growing interest in the use of technology, including machine learning methods, to promote human collective reason, including democratic deliberation [21, 13]. More traditional ML approaches (e.g., data/opinion mining) make use of richly structured (e.g., graph-based) models to make sense of public discussions [36] and to facilitate interactions in online communities [16]. We seek to leverage recent breakthroughs in large-scale language modelling to facilitate public deliberation via consensus generation.

## 3 Methods

We created a large data set of debate questions and built a customized environment and pipeline that allowed us to collect human opinions and fine-tune our models in an iterative loop (Figure 1).

### 3.1 Generating debate questions

We generate questions using a prompted, 70 billion parameter pre-trained LLM (*Chinchilla* [17]). We seeded the process with 152 hand-written questions on issues of contemporary debate. Most were policy questions of the form "Should we...?" or "Should the government...?" that are relevant for the UK-based participants that took part in our human evaluation. We use these 152 seed questions to artificially generate a total of 3500 debate questions. For each debate question, we prompt *Chinchilla* with a sample of 10 seed questions. The model generates a new question, which, if unique, we add to our total set. We manually check questions and filter out any that we consider likely to elicit extremist views or discriminatory language from the users.

This results in 2922 questions which we use to create a training set and two test question sets by clustering according to topic. We first embed each question using a Universal Sentence Encoder and cluster the questions into 110 topics using k-means clustering [9]. There is, for example, a cluster of 25 questions that corresponds to questions on food taxes such as "Should there be a tax on junk food?"

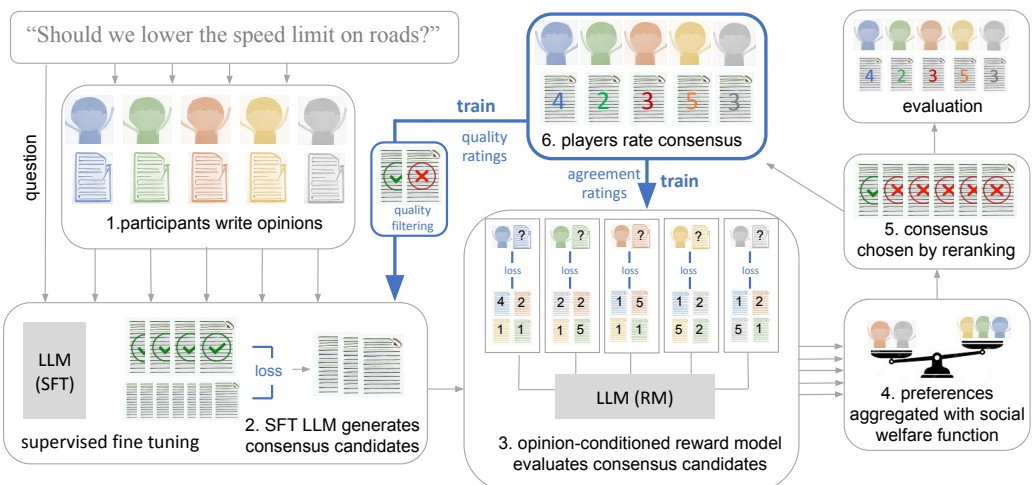

Figure 1: Overview of the data collection procedure. The evaluation pipeline proceeded in six steps. (1) Human participants, sorted into small groups ($n \in \{3, 4, 5\}$), each wrote a short paragraph stating their opinion about a political question (e.g., "should we lower the speed limit on roads?"). (2) These opinions, together with the question, were passed to a prompted pre-trained LLM (or, a fine-tuned LLM on later rounds) via the prompt, which generated consensus candidates. (3) Pairs of participant opinions and candidate consensus statements were passed into a reward model, which estimated the degree to which each participant would agree with a candidate consensus. (4) For each consensus candidate, the set of predicted individual preferences were aggregated with a social welfare function. (5) From a batch of consensus candidates, the one that maximised welfare was selected for human evaluation. (6) Participants then rated this consensus candidate, together with candidates generated in other batches or conditions, on a 7-point agreement scale. Quality ratings were used to filter the data for later fine-tuning and agreement ratings for training the reward model and for evaluation.

and "Should we remove all tax on food and groceries?". We then split the clusters into two groups, and set some question-clusters aside for an out-of-distribution hold out set (n = 302). The remaining questions were sorted into a training (n = 2320 questions) and a within-distribution hold out set (n = 300). See Appendix A for more information on question generation, filtering and clustering.

## 3.2 Data collection and environment design

On each round, groups of UK-based participants ($n = 3211$ organized into 746 groups, combined across training and eval) viewed a question, and wrote their opinion by typing freely into a text box in our custom online application. The opinions of a group were put into a prompt that was provided to one or more LLMs to generate candidate consensus statements. Data was collected using groups of four or five participants, though candidate consensus statements were sometimes generated based on a subset of participants' opinions (Section 4.3). Consensus candidates were then presented individually to participants in a random order, and each participant rated them along two dimensions, quality and agreement, using 7-point Likert scales. Participants rated all candidate consensus statements twice, to allow us to measure intra-rater reliability.

Additionally, before providing their own opinion or viewing candidate consensus statements, participants provided an agreement rating on a "position statement", which is a version of the question stated declaratively (e.g., "We should..."). This allowed us to measure baseline disagreement among the group. Each training or evaluation session took between 45 minutes and an hour. As our explicit goal is to train and evaluate on diverse opinions, we recruited a new set of participants for each data collection session, rather than use the same participants repeatedly. The full details of our study design, including compensation rates, were reviewed by our independent ethical review committee. All participants provided informed consent prior to completing tasks and were reimbursed for their time. It is our policy that researchers must pay workers/participants at least the living wage for their location. For this study participants were paid an average compensation rate of £15 per hour (the total cost of the study was approximately £46,000). No personally identifiable information was collected

as part of this research, nor was any offensive content shown to participants. More detail on the environmental design and the recruiting and training processes can be found in Appendix B.

## 3.3 Group alignment

In line with previous work [37, 30, 26], we use reward models to predict whether generated statements will be preferred by participants. Given our focus on diverse preferences, we train a reward model that predicts agreement conditional on a person's own opinion. Given a group of people with corresponding opinions, we can then use this model to generate, for each person, a score that predicts how likely they are to agree with a given statement.

Having estimated each individual's expected agreement, we aggregate these scores to predict the extent to which a candidate consensus will be preferred by the group. For simplicity, we assume that preferences are cardinal and comparable between participants, and we aggregate them using a cardinal Social Welfare Function (SWF). SWFs are used in the field of welfare economics to map a set of numeric individual utitilities to collective welfare [19]. The goal of the model is then to generate a consensus that maximizes the group welfare given the set of opinions from the group. Any SWF that satisfies six desirable axioms belongs to a one-parameter family of isoelastic social welfare functions [24]

$$
W_\alpha(u_1, \ldots, u_n) = \begin{cases} \left[ \frac{1}{n} \sum_{i=1}^n u_i^{1-\alpha} \right]^{\frac{1}{1-\alpha}} & \text{if } \alpha \geq 0, \alpha \neq 1 \\ \sqrt[n]{\prod_{i=1}^n u_i} & \text{if } \alpha = 1 \end{cases}
\tag{1}
$$

where $u_i$ is the utility of person $i$ and $\alpha$ is the degree of inequality aversion. At one extreme, for $\alpha = 0$, the SWF corresponds to max-mean or Utilitarian, which computes the mean expected agreement across the group. On the other extreme, for $\alpha = \infty$, the SWF corresponds to max-min or Rawlsian, which maximises the expected agreement for the most dissenting group member. Rawls argued in favour of the welfare function that yields the most desirable condition for the worst-off member of the group [28]. In the Results section we use a Utilitarian social welfare function for ease of interpretation but we present results on Rawlsian and Bernoulli-Nash (max-product, $\alpha = 1$), in Appendix D.5.

## 3.4 Training

Our training pipeline largely follows previous reports that use human feedback to fine-tune large language models [37].[2] Thus, we use a supervised fine-tuned LLM to generate $N$ consensus candidates ($N = 16$), which are then reranked by a reward model according to their expected social welfare. We then select the statement that maximizes the welfare. We denote these models SFT-@SWF where @SWF corresponds to welfare function.

The reranking approach gives us flexibility over the social welfare function during deployment.[3] To ensure that the model generalizes to different social welfare functions on the Utilitarian-Rawlsian axis, we sample the inequality aversion parameter $\alpha$ during training time from a log-normal distribution. During training, we start with the 70 billion parameter pretrained Chinchilla model and iterate twice over the following training steps before evaluating our final models.

**Step 1 - Generate consensus candidates and have them rated by humans** Participants provide written opinions in response to a question. The fine-tuned language model takes in the question and these opinions as part of its prompt and generates a set of consensus candidates (on the first iteration, we bootstrap first with zero-shot prompting and then with few-shot prompting of the base *Chinchilla* model). To ensure that our dataset contains data on a variable number of opinions (between 3 and 5), each time a statement was generated we silently omitted 0, 1, or 2 of the participants' opinions. For each unique number of opinions, we generate 16 candidates and select 2, each of which is ranked top-1 under a different $\alpha$. These 6 (3x2) highest ranked consensuses are then presented to participants who rate them for quality and agreement using two 7-point Likert scales. Note that, on the first

---

[2]Supervised fine-tuning with demonstrations from expert human consensus-writers (with access to the opinions) could be a viable alternative. However, previous work on summarisation has shown that learning from demonstrations can result in the model prioritising the fine-tuning objective (maximising the likelihood of a demonstration, including low-quality examples) over the true objective (in this work maximising welfare) [30].

[3]Recent work has shown that reranking can perform on-par or better than directly optimizing a model to maximize human preferences using reinforcement learning [23, 32].

iteration, we have no reward model for the reranking and selection scheme and thus simply generate 2 candidates for each unique set of opinions. Across all training data collection runs, 1524 participants contributed to our training data. See Appendix B.1 for more details.

**Step 2 - Supervised fine-tuning (SFT) to improve quality** We fine-tune a pretrained *Chinchilla* model on the consensus candidates that were rated as high quality (mean quality of 6 or higher on a 7-point Likert scale). The purpose of SFT is to familiarize the model with the prompt template and increase the candidate quality. We do not use agreement ratings to filter candidates as we aim to retain diversity in the kinds of stances expressed in the candidates so we can then use reranking to find the best candidate in terms of welfare[4]. SFT training details including the prompt template and hyperparameters can be found in Appendix C.1.2.

**Step 3 - Train a reward model (RM) to predict preferences** We train an RM to take in a question, opinion and a statement, and output a scalar "agreement" score. The score is a proxy for how likely an individual is to agree with a statement given their own opinion. To provide data for model training, each participant rates the six consensus candidates along a 7-point Likert score. Note that, as we generate candidates based on 3, 4, and 5 opinions, participants also rate candidates that were generated without taking their opinion into account. We map these six ratings to $\binom{6}{2}$ pairwise comparisons, remove the rating ties, and remove ratings from participants with low intra-rater reliability (see Appendix B.2.1). The RM is then trained to predict which statement out of a pair each user will prefer, conditional on the question and opinion, using standard cross-entropy loss. Note that the RM only conditions on one opinion, while the SFT conditions on all opinions. We warm-start the RM using a pretrained *Chinchilla* model and add an extra final linear layer to predict the reward. The prompt template and further training details can be found in Appendix C.2. After training the reward model, we go back to step 1.

# 4 Results

We ran two human evaluations, comparing our model against both high-quality baselines and human-generated opinions. These evaluations used the same data collection platform and basic design (see Section 3.2) and were run in separate sessions using the within-distribution questions and out-of-distribution questions. We ran an additional evaluation experiment to assess the model's sensitivity to different social welfare functions, but this did not reveal any reliable differences in the average or minimum agreement rating under any of the welfare functions of theoretical interest (we discuss this null result further in Appendix D.5).

## 4.1 Preferences for consensus over baselines

We first test our main model against a set of high-quality baseline models. Our main model (**SFT-Utilitarian**) generates statements using a 70 billion parameter language model fine-tuned on high quality consensus statements. At inference time, we sample 16 statements based on the question and opinions provided, and select the statement that maximizes predicted welfare under a Utilitarian (max-mean) aggregation function. The baseline models are:

- **SFT-Base**. Our fine-tuned model but without the aggregation function selection process, sampling only one statement at inference time.

- **Few-shot**. A few-shot prompted *Chinchilla* model. Each prompt contains three real examples composed of consensus statements with three, four and five opinions. Each set of examples is sampled from data collected using a zero-shot model, using a combination of quality and agreement criterion (see Appendix C.1.1 for more details).

- **Zero-shot** A prompted *Chinchilla* model without examples.

Under each of the four model types, we generated two different candidates, all of which were presented to the participants for rating. In these evaluations against baseline models we examine and compare performance of the model under within-distribution questions (collected using groups of

---

[4]Our SFT approach is similar to RL via Expert Iteration [2], alternating between (1) policy improvement, sampling from the current model and filtering these samples for quality, and (2) distillation, training a new model using these filtered samples.

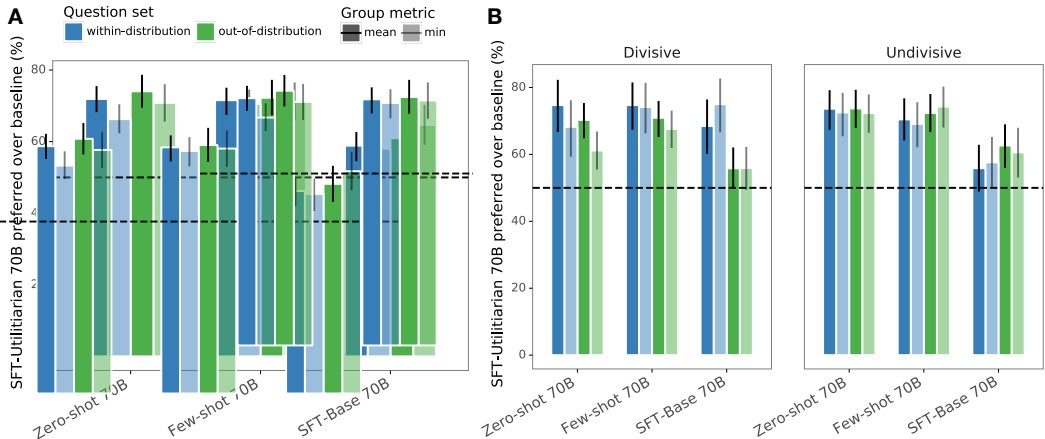

Figure 2: Win rates for comparing models constructed by pairwise comparison of Likert agreement ratings for candidate consensus statements (excluding ties) for within-distribution (blue) and out-of-distribution (green) question sets. Likert agreement ratings are aggregated within groups by either the mean (dark bars) or the minimum (light bars) agreement score. A: Win-rates for the SFT-Utilitarian model in comparison to baselines. B: Win-rates for the SFT-Utilitarian model broken down by whether or not the question was divisive in the group (see main text for details). Error-bars represent 95% bootstrapped confidence intervals.

five participants; total $n = 530$), and out-of-distribution questions (collected using groups of four participants; total $n = 267$).

We found that the position statements on approximately 50% of rounds (questions addressed by particular groups) were *undivisive* (receiving either all agreement or all disagreement from the group, when examined in a binary fashion; see Appendix D.6); we perform a split of the dataset into *divisive* and *undivisive* questions for further analysis. Notably, the fact that 50% of questions contained at least one dissenting participant (out of a group of up to five people) indicates that our population of participants do indeed have divergent opinions on many topics.

We first compare how likely a group is to prefer one policy over another. We do so using two metrics: the mean agreement across the group (consistent with the Utilitarian training objective of the model) and the minimum agreement in the group (consistent with a Rawlsian objective, not explicitly used in the model).[5] We compute these metrics for each statement and take the mean metric value across the statements generated under the same policy. To compute the win-rate, we compare these mean scores between two policies where the "winning" policy is the one with higher score. Under the group-mean agreement score, participants significantly prefer our main *SFT-Utilitarian* model over all of the baselines (Figure 2A), and strongly so in comparison to the two prompted LLMs. Even more impressive, participants prefer the *SFT-Utilitarian* model over all baselines for the group-minimum agreement score (a Rawlsian objective), indicating that our model is also more adept at increasing agreement among the strongest dissenters. Furthermore, we see the human preference for our model's consensus statements is present for both more and less divisive questions (Figure 2B), highlighting the model's utility in helping people find agreement even where opinions are divided.

In addition to the main baseline model comparisons detailed here, we also compared our main model against a smaller 1.4B parameter model ($n = 224$). We found that training a smaller model with data generated from the larger model can be effective, as the *SFT-Utilitarian-1.4B* outperforms the larger prompted models. However, the larger SFT-Utilitarian-70B is still superior, indicating that fine-tuning and size are both additive in this task. See Appendix D.1 for full details.

---

[5]We additionally analyze the group agreement scores in terms of the median, which is more appropriate for an ordinal (Likert) scale, and is more robust to outliers in small samples, as we have here in our groups. Usage of the median results in the same substantive conclusions for the primary analyses and for clarity we report these results in Appendix D.3.

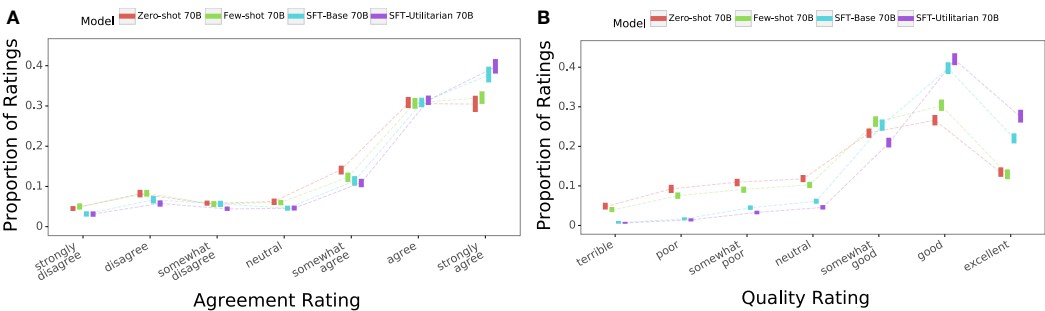

Figure 3: Distributions over Likert ratings for candidate consensus statements generated by the SFT-Utilitarian model and baseline models. A: Agreement ratings. B: Quality ratings. Error-bars represent 95% bootstrapped confidence intervals. See Figure A11 for agreement scores broken down by question divisiveness.

To further assess the reliability of the human preference for our *SFT-Utilitarian* model's consensus generations over those of the ablated models, we constructed a maximal mixed-effects logistic generalized linear regression model [22], the standard in confirmatory hypothesis testing [6]. Consistent with our win-rate analysis, we found that participants agreed more strongly with the consensus candidates generated by the *SFT-Utilitarian model* than those generated by the *SFT-Base* ($\beta = 0.12$; SE = 0.047; z = 2.53; $p = 0.011$; Figure 3A). Under this analysis, participants also preferred the *SFT-Base*'s generations more so than those of the *few-shot* model ($\beta = 0.40$; SE = 0.055; z = 7.35; $p < 0.001$), but exhibited no preference for the *few-shot* model over the *zero-shot* model ($\beta = 0.05$; SE = 0.06; z = 0.88; $p = 0.34$). We observe that the quality of the model-generated statements also increases as a function of our training pipeline, with both supervised fine-tuning and reward modelling impacting participants' quality ratings (Figure 3B).

To assess the performance of the SFT-Utilitarian model in more absolute terms, we examined the data for the approximately 50% of rounds in which participants were on different sides of a position statement (i.e., the *divisive* rounds). We found that, on these divisive rounds, 65.6% [61.9, 69.3] of candidates generated by the SFT-Utilitarian model were less divisive than the initial position statements. Furthermore, on 40.8% [35.4, 46.2] of rounds with a divisive position statement, a candidate consensus statement from the SFT-Utilitarian model achieved unanimous support (i.e., all participants somewhat agreed with the statement; see Appendix D.7 for full results and details). These results demonstrate not only that human participants preferred the consensus candidates generated by the SFT-Utilitarian model over those of the ablated models, but that the SFT-Utilitarian is demonstrably effective at helping a group of people with diverse preferences find points of agreement.

## 4.2 Preferences for model candidates over human opinions

Here we compare the performance of our *SFT-Utilitarian* model against human-generated opinions. While the human opinions are not explicitly written with consensus in mind, the opinions are very high-quality, frequently containing well-reasoned justifications for their positions (see Tables A6-7 for examples). Data was collected using groups of four participants ($n = 189$ for within-distribution questions, $n = 186$ for out-of-distribution questions). As with our previous evaluations, each participant wrote an opinion in response to the question, and these opinions were used to generate three candidates with our *SFT-Utilitarian* model. The same participants were then shown these model-generated consensus candidates alongside the (anonymous) opinions of the other participants in their group. All participants rated both the consensus candidates and the opinions.

We perform two analyses: for each participant, we compare their mean agreement over the set of other participants' opinions to their mean agreement over the set of model candidates; we also compare their most preferred other opinion to their most preferred candidate. We then compute the win rates over all participants and questions for each of these two metrics separately. The mean candidate score is preferred over the mean opinion score 78% (95%-bootstrapped CI: [75%, 80%]) of the time. Perhaps more impressively given the potential variance across the different opinions, even when we select the best-rated opinion, we find that our best-rated model candidate is still preferred 65%[61%, 69%]

of the time. Finally, we find that this difference in preference is larger for more divisive questions (win-rate of $66\%[61\%, 71\%]$) than for less divisive questions (win-rate of $63\%[57\%, 69\%]$).

## 4.3 Opinion exclusion analysis

Our model's ability to generate statements that provoke higher agreement could be achieved by generating statements that are more likely to be preferred *in general* but not tailored to the specific set of opinions of the small group. In order to assess this potential failure mode, we make use of the fact that our main evaluation dataset contains candidates that are based on subsets of opinions (3 or 4) from the full set of five provided by the participant group. If the model is making use of the specific opinions passed into it, then the agreement rates from participants whose opinions are excluded from the candidate generation process should be lower than those whose opinions are included. For each candidate in our evaluation set where fewer than 5 opinions were included, we compute the difference in median agreement between the inclusion group and the exclusion group. This analysis reveals an average inclusion/exclusion Likert agreement difference of $0.47$ (CI = $[0.21, 0.73]$) for our *SFT-Utilitarian* model. Quality ratings, by contrast, are not impacted by opinion exclusion (Mean= $0.10$, CI= $[-0.05, 0.25]$), and the difference between quality and agreement scores under this exclusion analysis is statistically significant ($t = 2.94, p = 0.0036$). This dissociation supports the conclusion that our model is successfully producing consensus candidates based on the specific set of opinions provided by the users, rather than producing generically preferred statements.

## 4.4 Out-of-distribution generalisation

So far, our evaluations are based on a question set that was unseen during training but that came from the same topic clusters of questions that were used during training. Next, we run separate human baseline experiments using the out-of-distribution question dataset, which has been specifically created so that the questions came from topic-space clusters that were never seen by our models during training. Despite the topic novelty, we find that that the *SFT-Utilitarian* model's statements are preferred over all baselines to a degree that is numerically similar to that for the within-distribution questions (Figure 2). *SFT-Utilitarian* model also continues to outperform the human opinions in the out-of-distribution question set at similar rates to the within-distribution question (mean candidate vs. mean human opinion: 76% [74%, 79%]; max candidate vs. max human opinion: 59.5% [56%, 63%]). Overall, this additional set of evaluations demonstrates the capability of our model to generalize beyond its training distribution without apparent loss of performance.

# 5 Discussion

We fine-tune an LLM to take in a question and the written opinions of a human group, and generate a statement that maximises the agreement of that group. This work opens up new avenues for language modelling in which the goal is to accommodate a set of diverse preferences.

## 5.1 Limitations

**What makes a good consensus?** Consensus statements generated by our model are rated more highly than those produced under rival methods, including one baseline (SFT-Base) that mimics our approach in every respect, omitting only the reranking step. Moreover, the model is sensitive to the specific opinions provided in the prompt, because auxiliary analyses show that excluded participants offer lower agreement but similar quality ratings. However, we do not know exactly *why* the statements are preferred. In particular, SFT-Utilitarian also receives a higher number of "excellent quality" ratings (see Figure 3B) raising the possibility that its statements (whist tailored to the group) are also more generally sensible or well-written. Alternatively, people may use "quality" as a proxy for agreement, raising the additional risk that more confident or authoritative users will wield more influence over the consensus, potentially exacerbating power imbalances between different social groups.

**Social Welfare Functions.** We train the model under a distribution of SWFs. In the Results section we show that reranking statements using a Utilitarian aggregation function generates consensus statements that already take into account both minority and majority views (improving both the min and mean rating over baselines). Hence, when we compare the Utilitarian SWF to other more equitable welfare functions like Rawlsian and Bernoulli-Nash, we do not observe meaningful

differences in the mean ratings across participants or that of the most dissenting (min) participant. We thus cannot conclude based on these results alone that aggregating using different SWFs actually results in different behavior. We report these results in Appendix D.5.

**Data collection** We collect human data from a crowd-sourcing platform. Because the debate questions are relevant to UK current affairs, we limit inclusion to UK participants. However, this curtails the diversity in our participant cohort, and limits the generalisability of our findings. It also raises the risk that consensus statements may unduly reflect the views or biases of the participant demographic that we have sampled, in addition to any biases that may arise during pretraining [34, 7]. We note that the benefits of our model are most pronounced at the upper end of the Likert scale, perhaps because homogeneity in our sample inflates baseline levels of agreement. We are developing additional recruitment methods to allow us to sample a more diverse group of people for future experiments.

**Scale** We limit data collection to small groups of four or five people. This choice is in part because of the technical limitation imposed by the prompt length of the model, which can handle only a handful of written opinions. There may, however, be important future use cases where aggregating over many thousands of opinions is necessary, requiring an architecture that is scale invariant. One approach that could be fruitful is to map each opinion to an embedding and then aggregate those embeddings directly. Related examples include the attention-based architecture in [33] or the recursive approach developed for summarizing books in [35].

## 5.2 Broader Impacts

**Misuse for persuasion** We did not train the language model to adopt a particular position or persuade others of a specific political view. Nevertheless, there is a risk that LLMs can be used for human persuasion, posing a risk in political discourse, the media, or in advertising. Political debate is already increasingly polarised, especially in Western societies. A system that is capable of persuading others to adopt a particular viewpoint could learn to present arguments in a manipulative or coercive manner, and mitigations for these potential harms play an important part in any research addressing this topic.

**Factuality** The language model we describe is not specifically fine-tuned to produce consensus opinions that are factually accurate. Thus, whilst manual review of consensus statements suggested that they were broadly accurate, there exists a risk that the consensus opinions that it produces could be misleading or contain false information.

**Misrepresentation of consensus** Our work describes an AI that helps people find agreement in natural language. However, the consensus statement might not reflect the views of all the users. This raises important questions about how such a consensus is used. For example, a consensus might be presented as reflecting a unanimous view, misrepresenting the minority opinion, or use the consensus statement to justify otherwise unwarranted courses of action. It is important that users understand these caveats when interpreting the consensus.

**Opportunities** Nevertheless, despite these acknowledged risks, our research was conducted with societal benefit firmly in mind. The ultimate goal of our work is to provide a tool that can be used safely to help people find agreement. We focus on opinions about debate questions, but we can envisage a wider set of use cases, such as aggregation of online reviews into more helpful meta-reviews, systems for collective writing that automatically takes the preferences of different authors into account, and systems for collective decision making for organised groups. However, we note that considerable work is needed to understand the potential risks associated with AI consensus generation, and to find ways to ensure that model outputs are generated in a transparent and explainable way, before any such system can be deployed.

## Acknowledgements

We would first like to thank the human participants, whose high-quality opinions and ratings allowed us to train and evaluate our models. We would also like to thank Jonathan Uesato and Fan Yang for help with the LM infrastructure, Sarah Henderson, Richard Ives, Antonia Paterson and Jacklynn Stott for operational support, and Iason Gabriel, Will Hawkins, Geoffrey Irving, Raphael Koster, Angeliki Lazaridou, and Boxi Wu for helpful comments and suggestions.

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
