# Fine-tuning language models to find agreement among humans with diverse preferences

# Appendix

## Contents

36th Conference on Neural Information Processing Systems (NeurIPS 2022).

# A  Generating debate questions

## A.1  Generating questions from seed questions

We need access to a large and diverse dataset of debate questions to train and test our models. To generate these questions, we use the pre-trained *Chinchilla* model using few-shot prompting and 152 hand-written seed questions [4]. We go through the following process to generate each question:

1. We randomly sample 10 seed questions from the corpus of 152 hand-written seed questions. See Table S3 for 10 examples of seed questions.

2. We prompt the model with the 10 sampled seed questions, each delimited by a new line token '\n'. See the prompt template in Table S1. We use nucleus sampling with a cut-off probability of $0.8$ and a temperature of $1$ [5].

3. We then take the generated output, remove the `Question:` prefix and truncate everything after the first new line token.

4. If the generated question is longer than 15 characters, contains the word "should" and is not in our dataset yet, we add this question to our dataset of questions.

5. We repeat this process until we have 3500 questions in our dataset.

Note that the seed questions are not directly used in the final question set and that the newly generated questions are not used as additional seed questions.

## A.2  Curation process

To ensure that questions are safe and effective and can be used to elicit diverse opinions, we check each of the 3500 generated questions manually and exclude questions for any of the following seven reasons:

1. The question is nonsensical or logically incongruent.

2. The question requires the reader to understand technical jargon unlikely to be in the public vocabulary.

3. We excluded questions that we felt might be likely to provoke extremist views or discriminatory attitudes, in particular where the discussion would be at risk of differentially impacting protected groups, including age, gender reassignment, being married or in a civil partnership, being pregnant or on maternity leave, disability, race including colour, nationality, ethnic or national origin, religion or belief, sex, and sexual orientation.

4. The question proposes violence against someone or a group of individuals.

5. The question refers to decisions that other countries or populations should make and hence is not targeted at our target demographic (UK citizens).

6. The question will not lead to a diverse discussion by virtue of being either too specific or too broad to have reasonable disagreement.

7. The question contains one or more false presuppositions.

After curation, there were 2922 questions remaining.

## A.3  Topic clustering and out-of-distribution question set

We split the remaining 2922 questions by topic using k-means clustering, and allocate some clusters to a special out-of-distribution test set. In this way we can test how our model generalises to topics that are unseen at training time. To identify the topic of each question, we use a Universal Sentence Encoder [2] with k-means clustering and follow the steps below:

1. First, remove phrases that occur in many questions and are irrelevant for clustering into topics. This includes the following strings: "should we", "should the government", "government", "the uk", "the united kingdom", "britain", and "great britain".

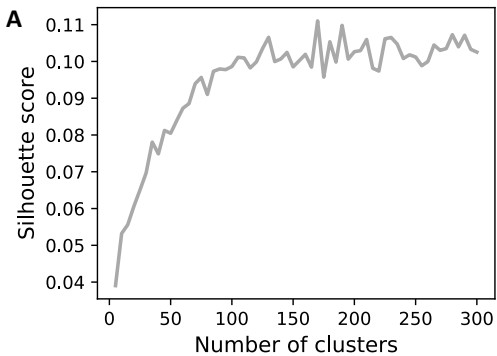
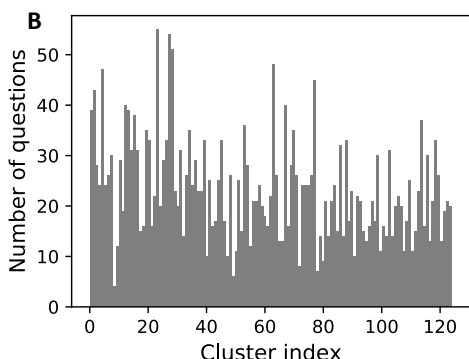

Figure S1: K-means clustering of the debate question embeddings. A: The Silhouette score as a function of the number of clusters. B: The distribution of questions per cluster (total of 2922 questions).

2. We encode each question using a Universal Sentence Encoder (USE) via Tensforlow Hub (using the following module: `https://tfhub.dev/google/universal-sentence-encoder/4`). This maps each question to a 512-dimensional embedding.

3. Using the standard `sklearn` k-means implementation, we cluster the USE embeddings while sweeping the number of total clusters from 5 to 300 in increments of 5. For each number of clusters, we randomly reinitialize 100 different times, and keep the clustering that that produces the highest Silhouette score. We find that the Silhouette score converges at 125 clusters (Figure S1A). We use these 125 clusters for choosing train and test question sets and show the distribution of questions over these clusters in Figure S1B.

4. We then randomly allocate some clusters one-by-one to an out-of-distribution question set, and continue adding clusters until we have at least 300 examples in this set. To ensure similarity of topic-spread between the within-distribution and out-of-distribution question sets, we create 1000 random splits between the within-distribution and out-of-distribution set and measure the mean Euclidean distance between clusters in both sets. We then choose the within-distribution and out-of-distribution sets as those that have the most similar mean cluster distance. The final out-of-distribution test set has 302 questions across 14 different topics.

We refer to Table S2 for example questions from each a subset of clusters. Finally, the within-distribution question set is split into a training set and a within-distribution test set, which contains questions which are different from the training set but belong to the same clusters.

## B  Human data collection

### B.1  Experimental protocol

All participants were recruited via an online crowd-sourcing platform, and took part in our experiments via a web interface. Human data was collected for both training and evaluation using the same basic experimental protocol and custom interface, which we describe here in detail. Each participant first read the task instructions (see Figure S2), and completed a short comprehension test. The comprehension check was designed to test the participants' knowledge and understanding of key aspects of the experiment. Each question was multiple choice, and after an answer was selected we provided immediate feedback to consolidate or correct the participant's understanding. We did not exclude any participants at this point. Upon completion of the comprehension test, participants were redirected to the experiment lobby, where they were asked to wait until enough other players had joined to form one experimental group (the size of which was always either four or five, depending on the particular experiment - details are in the main text). Once all players had joined, the group started the main experiment. In practice, data was collected in batches of around 20 groups (100 participants) in parallel.

Each experiment consisted of four rounds, each of which was based on a different political question. Within each round, there were three consecutive phases: opinion writing, candidate generation, and candidate rating. First, participants would view the question of interest, and would be provided with a text box in which they were asked to write their own opinion in response to the question. Participants were explicitly asked to write their own opinion on the topic, and not to disengage from the experiment in order to read any external sources. Pasting was disabled in the text box to further discourage the use of external opinions. Opinions had to be between 10 and 200 words in length and participants were encouraged to write between 3 and 10 sentences. As this was a multiplayer game, we applied a time limit of 5 minutes to the opinion writing to ensure that participants were not forced to wait too long for others to finish. The time remaining was always displayed at the top of the page during this phase. If a participant failed to complete within this time limit, the game moved on to the next phase, and their opinion was not included in the candidate generation process. Once all participants in a group had submitted their written opinions, these were passed to one or more models, and used to generate a set of consensus candidates. During this phase participants' viewed a waiting page which could last between 20 to 120 seconds depending on the particular experiment.

Once the candidates had been generated, participants progressed to the final phase - candidate rating. The user interface for this phase is displayed in Figure S3. Both the question and the participant's own written opinion were displayed as a reminder, and the consensus candidate was displayed beneath. Two drop-down menus allowed participants to provide seven-point Likert scores for both agreement and quality (full labels are provided in Table S4). Prior to rating the model or human-generated candidates, participants were asked to provide an agreement (but not quality) rating to a 'position statement', which was simply the question itself restated as a declarative statement (e.g. The position statement for the question 'Should we lower the speed limit on roads?' was 'We should lower the speed limit on roads.'). This initial rating provided us with a general indication of the range of opinions to the question, independent of the candidate generation process. We used these statements primarily for determining the divisiveness of each question (see main text and section below).

Following the position statement, all consensus candidates were presented sequentially. All were presented once in random order, then again in a new random order. This repetition of each statement allowed us to measure the intra-rater reliability, which we used for data quality filtering (see Section B.2.1). Our intention here was to filter out ratings from participants that often varied hugely between their first and second ratings, as we consider this to be a proxy for the participant paying attention to the task (rather than responding randomly). We expected participants' two responses to be roughly the same, with some tolerance for noise, which informed our usage of a relatively lenient intra-rater reliability threshold. As with the opinion writing phase, participants were given a time limit to complete all of the ratings during the ratings phase. The precise limit varied between experiments, as different experiments presented different numbers of candidates. In all cases, the time limit was based on allowing approximately 60s per candidate. Time remaining was displayed at the top of the page throughout this phase, along with additional information regarding the number of questions remaining, and candidates left to rate. If a participant ran out of time more than once, we removed them from the experiment and they were still compensated at the full rate.

Additionally, we provided a button labelled 'Report Harmful Content', which participants were instructed they should press if they were exposed to any content that they deemed to be offensive or inappropriate. In the event that this button was pressed, participants would be taken out of the experiment and would be asked to (optionally) provide us with further information about the harmful content via a text box. Whenever this occurred, participants were compensated at the full rate for the experiment. However, we note that no participants reported any harmful content (full details in section below).

## B.2 Quality control processes

### B.2.1 Intra-rater reliability

Since participants provided two ratings (under each quality and agreement metric) for all model candidate consensus statements (and, human opinions, during the Human Opinion eval), we can compute an intra-rater reliability score for each participant. Intra-rater reliability provides a quantitative measure for how consistent participants' responses were across the two presentations of the same model statement (or human opinion). To measure intra-rater reliability, we computed Cohen's Kappa coefficient $\kappa$ using linear weights (i.e., linearly-weighted Cohen's Kappa), which allowed us to take

## Welcome to the Opinion Experiment!

This is a multiplayer game, where you and a group of other participants will all be asked questions about a variety of contentious topics. For each question, we would like each of you to write down your own, personal opinion on the topic. **This should include your opinion along with some justification for your view and any other information you think is relevant.** It is OK if you do not have a strong opinion - in this case please still tell us what you think about the statement, and any views you do have on the topic that might be relevant. The opinion that you write may be displayed to other participants once you have submitted it, but this will be completely anonymous. Nevertheless, please ensure that you do not write anything that could be considered harmful or offensive by others.

**Each opinion should be somewhere between 3-10 sentences (50 and 200 words) in length.** You will not be able to submit unless you have written at least 10 words, but please try to write at least 3-4 sentences. Additionally, the maximum word limit is 200 words. **We are interested in your own opinion, so please do not look up other information or text online while you are taking part in this experiment.**

Once you have submitted your opinion, you may have to wait for other participants to finish writing their own opinions. Please be patient during this time, as the experiment will continue shortly, and you may end up being timed out of the experiment if you do not pay attention (see more information on this below).

Once everyone has submitted their opinions, the opinions will be passed to an algorithm, which will produce one or more statements based on the combination of all participant opinions. Please note that this algorithm can take up to **two minutes to generate the statements**, so please be patient during this time. You will then be presented with a series of statements, one at a time. These may be a mixture of real human opinions (including your own), and algorithm-generated statements. You may also see a 'position statement', which is a single statement which takes a strong stance on the given question. For all of these statements we would like you to tell us how how much you agree or disagree with the statements, using a dropdown menu.

For all statements other than the position statement, we will also ask you to rate the general writing quality of the statement, regardless of how much you may agree with it. **A high quality statement should be clear, coherent, and should justify its stated position on the question.**

You will repeat this process for 5 different questions, after which the experiment will be complete.

Figure S2: Main task instructions provided to all participants. Additional information (not shown) was provided regarding the task time limits and the button for reporting any offensive or inappropriate content.

into account the fact that the ratings provided are along an ordered scale. In early batches of training data, we observed a bimodal distribution for $\kappa$, with a threshold of approximately 0.6 separating the "reliable participants" from the "unreliable participants" (0.6 is also commonly considered in the psychometric literature to be the threshold between "moderate" agreement and "substantial" agreement) [8]. Therefore, we used 0.6 as a threshold for filtering unreliable participants. Using this threshold, we retained 1028/1524 (67%) in our training data and 1164/1687 participants (69%) in the evaluation data. The empirical distributions over $\kappa$ for the full training and evaluation data sets are shown in Figure S4.

### B.2.2 Harm report

As described above, our user interface allowed participants to report any instances of offensive or inappropriate content being generated by the model, or indeed by other participants in conditions where participants viewed the opinions of others. An analysis of the data reveals that this functionality was used a total of nine times across all of our training and evaluation data collection, and in all cases this was explicitly reported to be a mistake by the participant. This indicates that over 746 separate groups of participants, our model did not generate content that was deemed sufficiently inappropriate or offensive to be flagged by the participants.

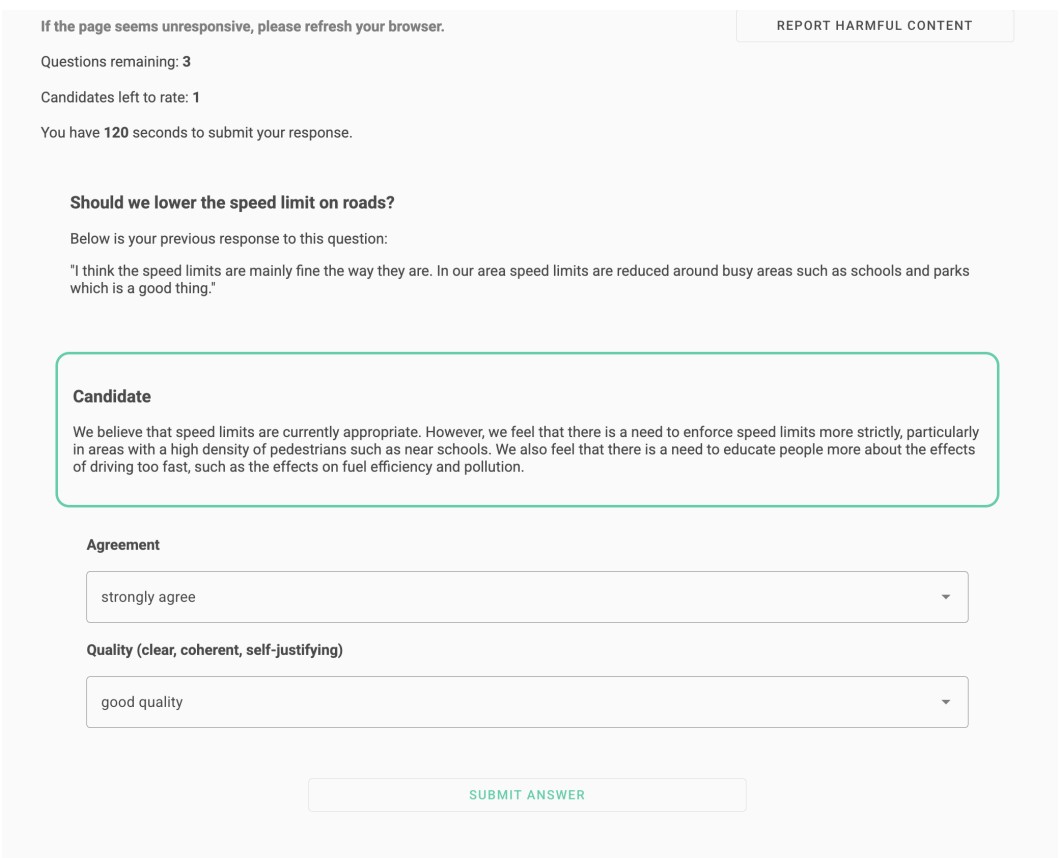

Figure S3: User interface displayed to participants when providing ratings for candidate consensus statements.

## C  Model training and evaluation

We use two different language models, a generative model for the consensus candidates and a classifier to predict the reward. The models are fine-tuned after each of the two consecutive rounds of data collection. The first round of data collection corresponds to participants that rate consensus candidates generated with zero-shot and few-shot prompting, while the second round corresponds to candidates generated using the models that were trained using the first round of data. For fine-tuning, we always start from a pre-trained *Chinchilla* model both for the consensus generating model and for the reward model. Moreover, after the second round of data collection we fine-tune our models using a mix of data from the first and second round, in line with prior work (e.g. [12]). Given the high cost of data collection and model training, as well as the relatively small increase in model performance between the first and second round of training, we stop data collection after two rounds.

### C.1  Consensus generating model

We use three different consensus generating models, a zero-shot and few-shot prompted *Chinchilla* model, and a fine-tuned (SFT) *Chinchilla* model. At inference time, we sample consensus candidates using nucleus sampling with a cut-off probability of $0.8$ and a temperature of $1$ [5]. When we use the reranking models (e.g. SFT-Utilitarian), we sample $16$ candidates and choose the sample that maximises the expected welfare. Note that the sampling happens live while participants wait for the candidate rating phase to begin. To increase candidate diversity and increase order-invariance, we use a different random order of opinions in the prompt each time we sample.

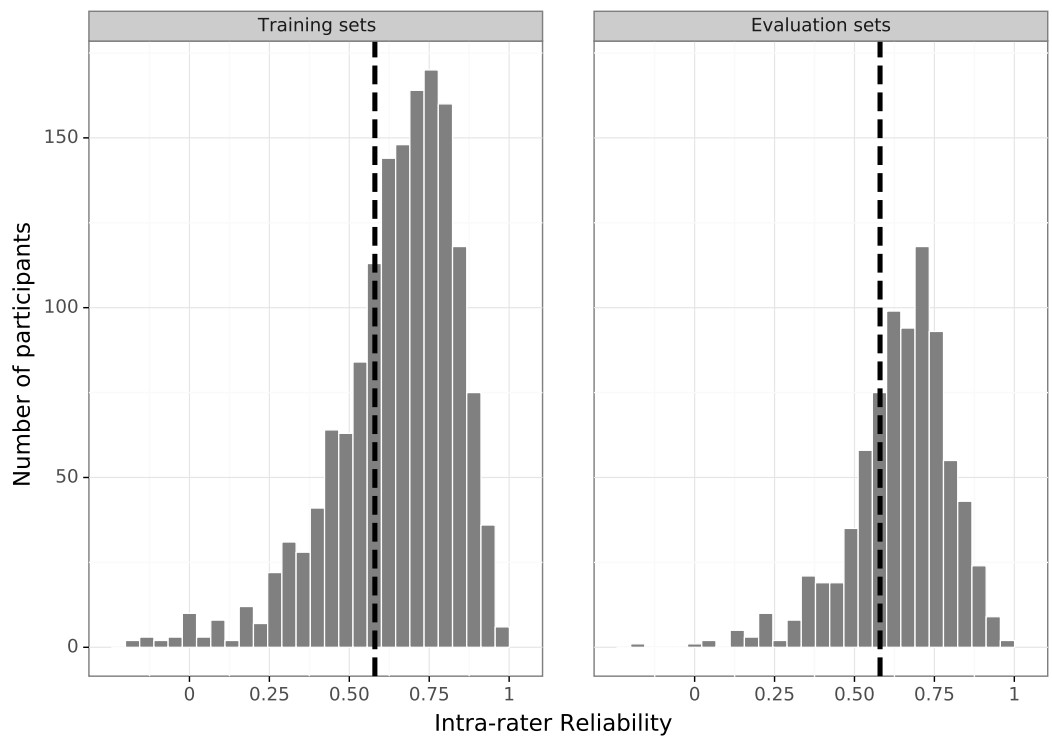

Figure S4: Intra-rater reliability using linearly-weighted Cohen's Kappa of Training data and Evaluation data. Vertical line shows our threshold for exclusion at $\kappa = 0.6$.

### C.1.1 Zero-shot and few-shot prompting

We use two approaches for prompting the base LLM model (*Chinchilla*). We refer to these two prompt methods as "zero-shot" and "few-shot" prompting. With zero-shot prompting, we simply provide a few sentences designed to elicit a prior based on the reports of citizens' juries, which we considered to be a good template for generating consensus statements. Following this, the prompt always included the specific question along with the human-provided opinions, but with no previous examples of what a consensus candidate statement should look like (see the zero-shot template in Table S1). For few-shot prompting, rather than create hand-crafted (and potentially unrepresentative) examples ourselves, we elected to collect some initial training data using the zero-shot approach. Once these initial batches were collected, we then created a generative few-shot prompting approach using real opinions and consensus candidates collected using the zero-shot prompting approach. By using real data for the examples in the prompt, the examples should be more representative of our participant population, while also allowing us to create sample diversity through our generative process.

The few-shot prompt was similar to the zero-shot one, but with the addition of three examples (see the few-shot template in Table S1). Each example consisted of a question, some opinions, and a model-generated consensus candidate statement, all of which were drawn from data collected using the zero-shot prompted model. Every prompt included one example with three opinions, one with four, and one with five, in order to encourage diversity and generalization over different numbers of input opinions. Each candidate generated under this method was based on a unique prompt generated using the following method, applied separately to the 3, 4, and 5 opinion candidates:

- Filter to include all examples where the candidates had a mean quality rating higher than "Neutral".

- Rank all consensus candidates under the specified social welfare function.

- Sample a candidate from the top 10 ranked candidates.

- Construct the example using the selected candidate along with the associated question and opinions.

This process provided a combination of prompt diversity along with a degree of example optimization towards the specified social welfare function in each case.

### C.1.2 Supervised fine-tuning

To further improve the consensus generating LLM over zero-shot and few-shot prompting, we fine-tune the pre-trained *Chinchilla* model using supervised finetuning, in line with previous works [9, 11, 12]. For supervised fine-tuning we use the same prompt as for zero-shot prompting, see Table S1. We also tried a simpler prompt without the first `A citizen's jury was tasked...` sentence but find that adding the sentence makes fine-tuning more data-efficient and more reliable.

**Training data**  While some previous works rely on human demonstrations for supervised fine-tuning (SFT), we find that many of the candidates that were generated using zero-shot and few-shot prompting are of sufficient quality to be used for further fine-tuning. Hence, we use highly rated consensus candidates from previous rounds (generated via prompting in the first round and generated with the first iteration SFT model in the second round) as target for fine-tuning using the following steps:

1. We first split the training data by question and allocate $80\%$ of the questions to a training set and $20\%$ to a validation set which we use for hyperparameter optimisation and early-stopping.

2. For the training data we then filter out the high-quality candidates as rated by our participants. These targets correspond to consensus candidates that have a mean rating higher than 5 (corresponding to "Somewhat Good Quality"), averaged over reliable participants (see Section B.2.1). For the validation set, we only use candidates that have a mean rating higher than 6 (corresponding to "Good Quality") to ensure that we select the highest-quality model. For training, we explored the quality-quantity trade-off by using different minimum mean rating thresholds ($\{4, 4.5, 5, 5.5, 6\}$) to build the training set but found that a threshold of 5 leads to the best performance on the validation set.

3. We further "clean" each candidate by 1) truncating any text after the first new line character 2) removing text that is repeated from the prompt (e.g. the model sometimes starts a consensus with `Opinion:`), 3) removing any leading or trailing quotation marks and whitespace.

4. Finally, to encourage order-invariance with respect to the opinions and increase diversity during the training process, we randomise the order of the opinions in the prompt at start of every new training epoch.

This process yields 1525 candidates for training in the first round of fine-tuning and 2355 candidates in the second round of fine-tuning. Note that in the second round, we use the examples from both rounds and thus use 3880 candidates in total.

**Training and evaluation**  During training, we evaluate our model performance using perplexity on the evaluation set, a measure corresponding to the inverse probability of predicting the test set using the current model. We stop training when the perplexity on the evaluation set no longer decreases. In both training rounds, we use a learning rate of $2.5 \cdot 10^{-7}$, a batch size of 256, and the AdamW optimiser [7]. To fit the 70 billion parameter model on TPU memory, we shard the model over 64 TPU v3 cores. We train all layers except the embedding layer and train in low precision using bfloat16 with stochastic rounding [9, 4].

In the first round, we train for 85 steps (14.3 epochs) and in the second round for 145 steps (9.6 epochs) before the model starts to overfit. Note that we train for more epochs than some other recent work that fine-tunes LLMs [9]. This can be explained by the fact that the order of the opinions in the prompt is randomised between epochs and the model thus rarely sees the exact same example multiple times during training. This also leads to a lower optimal learning rate compared to most previous work. We tried both higher and lower learning rates but found that $2.5 \cdot 10^{-7}$ yields the lowest evaluation perplexity.

In Figure S5A, we compare the perplexity for the SFT models to zero-shot and few-shot baselines using the data that was collected using the within-distribution and out-of-distribution test questions. We observe a performance increase both by going from prompting to SFT but also between both the two rounds of SFT.

## C.2   Reward model

In addition to the consensus generating model, we train a reward model that takes in the question, a single opinion and a consensus candidate and predicts a scalar reward. The inputs are combined using the prompt template for the reward model shown in Table S1. At inference time, we use the scalar rewards for each opinion to select the consensus candidate that is expected to maximise group welfare.

**Training data**   Following Christiano et al. [3], we train our reward model by predicting the preferred candidate using pairwise preferences. In our case, we extract those pairwise preferences from the agreement Likert ratings given by participants using the following process:

1. In line with SFT, we first split our data by question and allocate 80% of the questions to a training set and 20% to a validation set.

2. We extract all the possible pairwise preferences after removing ratings from unreliable raters.

3. We filter out all the rating pairs for which the difference in ratings is 1 point or less on the 7-point Likert scale. We also tried filtering out only strict ties (so allowing for example an ("Agree","Strongly Agree") pair) but find empirically that being stricter on which pairs we include leads to faster training and better evaluation performance, even when we use an evaluation set for which we filter out only strict ties.

4. Finally, before putting the consensus candidate in the prompt template that is fed to the reward model, we "clean" the candidate in a similar way as for SFT (see Section C.1.2).

This process yields 6639 rating pairs for reward model training in the first round and 3795 rating pairs in the second round. Note that, in line with SFT, we use the examples from both rounds (10434 pairs) for training the second iteration of the reward model.

**Training and evaluation**   At training time, we warm-start the reward model using the pre-trained *Chinchilla* model with an additional final linear layer to predict the reward. Some previous work has warm-started the RM with an SFT model. However, in our case the prompt templates are too different for this to benefit performance. Because we are predicting which candidate consensus is preferred out of a pair, we train and evaluate our models using a standard cross-entropy loss. We stop training when the evaluation loss no longer decreases. In both training rounds, we use a maximum learning rate of $2 \cdot 10^{-6}$ with a linear warmup and cosine anneal, a batch size of 8, and the Adam optimiser. To fit the 70 billion parameter model on TPUs, we shard the model over 32 TPU v3 cores. For the RM we only train the last 25 layers (including the extra linear layer) and, in line with SFT, we train in low precision using bfloat16 with stochastic rounding. To stabilise training during the beginning of training, we first freeze the pre-trained language model layers for 0.1 epochs and only train the extra linear layer during those steps.

In the first round, we train for 440 steps (0.5 epochs) and in the second round for 1113 steps (0.8 epochs) before the model starts overfitting. Note that, in contrast to SFT, we now train for less than one epoch which can be explained by the fact that we have extracted all possible combinations of pairwise ratings leading to many correlated pairs that share candidates. Hence, even though the model has not seen every unique rating pair during training, it has seen the vast majority of candidates multiple times.

To benchmark the performance of the final model, we compare in Figure S5b the pairwise accuracy (how often it selects the preferred candidate) for the final model, the reward model after the first training iteration, and six naive baselines:

- **Tf-idf with cosine similarity.** We train a Term Frequency Inverse Document Frequency (tf-idf) vectorizer on all the training data from both rounds and use this to vectorise the

opinions and candidates. We then compute the cosine similarity between the opinion and each of the candidates in a pair and select the candidate that maximises this similarity as the preferred candidate. We use the standard `sklearn` tf-idf implementation with English stopwords removed. To optimize the model, we sweep which n-grams are extracted (from only uni-grams to uni-grams, bi-grams and tri-grams), and we sweep the upper and lower thresholds for which n-grams to include based on how often they occur.

- **Choosing the longest.** We always select the candidate with the most characters as the preferred candidate, independent of the opinion or question.

- **ROUGE.** We always select the candidate with the highest ROUGE-1/ROUGE-2/ROUGE-L score [10]. ROUGE (Recall-Oriented Understudy for Gisting Evaluation) is a set of metrics for evaluating summaries and machine translation. We compute ROUGE by looking at the overlap of the candidates and the opinion. ROUGE-1 corresponds to unigram overlap, ROUGE-2 to bigram overlap, and ROUGE-L to the longest common subsequence.

- **BLEU.** We always select the candidate with the highest BLEU score [10]. BLEU (bilingual evaluation understudy) is a metric for evaluating machine translation. BLEU also computes the n-gram overlap but, in contrast to ROUGE, takes several size of n-grams into account simultaneously. Additionally, it introduces a brevity penalty.

For benchmarking, we use the data that was collected using the within-distribution and out-of-distribution test questions. In Figure S5b, we observe that both reward models outperform the naive baselines. Moreover, we observe a slight ($\sim 1\%$) increase in accuracy between the first and second round of reward model training.

## D  Additional human data evaluations

### D.1  Model size ablation experiment

In order to assess the impact of model size, we directly compared our main 70B parameter model against a smaller 1.4B model based on the same architecture and dataset as *Chinchilla* [4]. We fine-tuned this model and trained a 1.4B reward model using the data we previously collected with the larger model. We conducted a human evaluation experiment ($n = 224$), where we directly compared consensus candidates generated by 6 models: i) SFT-Utilitarian 70B, ii) Few-shot 70B, iii) Zero-shot 70B, iv) SFT-Utilitarian 1.4B, v) Few-shot 1.4B, and vi) Zero-shot 1.4B. As shown in Figure S6 SFT-Utilitarian 70B still significantly outperforms each baseline in win rate when comparing mean agreement. Interestingly, however, SFT-Utilitarian 1.4B performs only slightly worse (the 70B version wins over the 1.4B version 59.4% [53.7%, 64.7%] of the time). This effect size is similar to SFT-Utilitarian 70B versus SFT-Base 70B, suggesting that both scale and reward modelling are independently beneficial for consensus generation. Additionally, SFT-Utilitarian 1.4B outperforms both zero-shot and few-shot 70B (win rate of 75.9% and 75.2% respectively see Table S5). Notably, the smaller model was trained based on the data we previously generated using the 70B model. We speculate that the relative high-performance of the smaller fine-tuned model may be driven by the high-quality data generated by the larger model. Given the weaker prompting capabilities of the smaller model (see Table S5), it is likely that running the full iterative data collection pipeline from scratch using only the smaller model would have been more data intensive. This raises the interesting prospect of future, more efficient hybrid approaches, where initial data is generated using prompted large models, with fine-tuning applied to smaller models. Likert agreement rates for both small and large models are displayed in Figure S7, while example consensus candidates are provided in Table S7.

### D.2  Qualitative comparison of SFT-Utilitarian model to baselines

See an example in Table S6.

### D.3  Win-rate analysis on group median agreement scores

Here, we compare how likely a group is to prefer one policy over another by examining the median agreement scores for each group. The median aggregation function is a complementary statistic to analyze (in addition to the mean) for our data of Likert ratings, where we expect the resulting

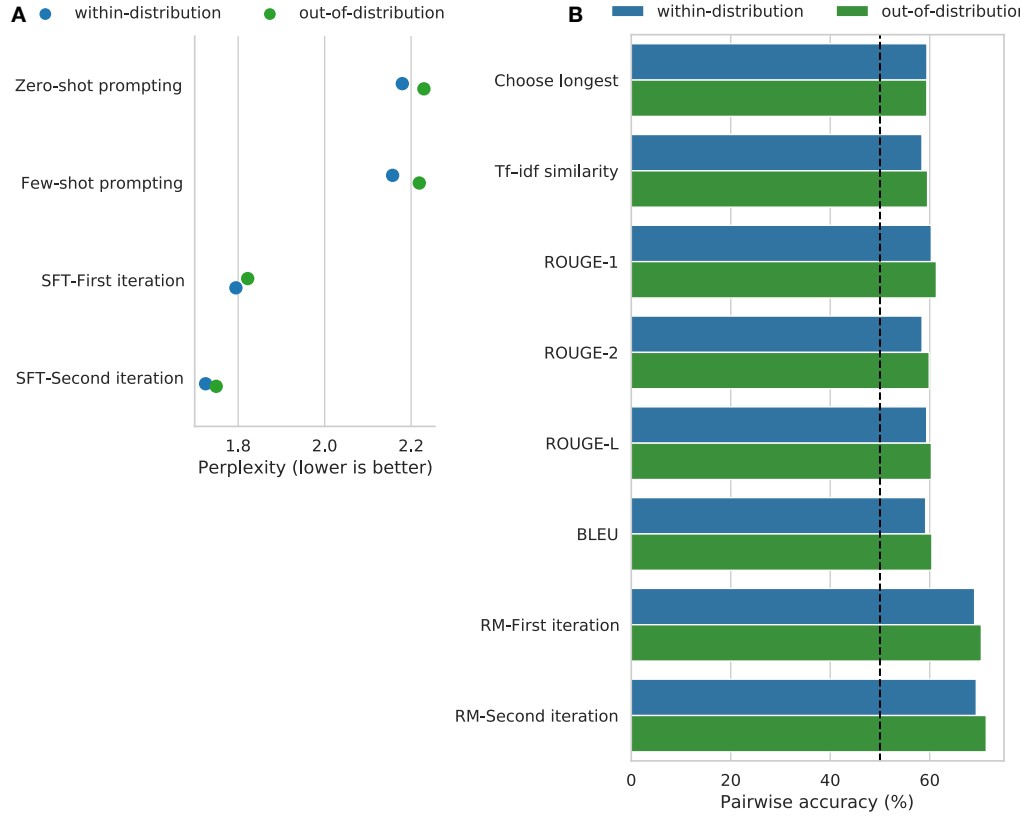

Figure S5: Model performance evaluated on the data that was collected for comparing SFT-Utilitarian to baseline models, using the within-distribution and out-of-distribution questions (see Sections 4.1 and 4.2 in the main paper). Given the size of the models, we train using only a single random seed and do not show error bars. A: Consensus generating model perplexity for the first and second SFT model compared to zero-shot and few-shot prompted. For this evaluation, only candidates with a mean rating higher than 6 were included in the evaluation set. B: Reward model performance in terms of pairwise accuracy (how well can the model predict which consensus candidate was preferred by a participant given their opinion and the question) compared to six baselines (see main text). For this evaluation, we use all the pairwise comparisons except when there is a strict rating tie.

distributions to be non-Gaussian (e.g., for our divisive questions, we expect bi-modal distributions). Further, for small sample sizes, the median is more robust to outliers. The substantive conclusions for our primary set of analyses are unchanged by examining the median (Figure S8).

### D.4 Regression analysis details

In the main text, we report the results of a mixed-effects logistic generalized linear regression model. This mixed-model allows us to take into account the fact that the individual ratings are not i.i.d., and instead structured in such a way that multiple ratings come from the same participant and that each question receives multiple ratings. We constructed a maximal random-effects model, with by-participant and by-question effects of intercept and slope (i.e., effect of model variant), which is the standard in confirmatory hypothesis testing [1]. In lmer-style syntax, the model formula was `Likert_rating` $\sim$ `model_variant + (1 + model_variant | participant) + (1 + model_variant | question)`. These random effects capture the idea that our various models might be better on some questions than on other and that different participants might show different levels of preference for some model variants over others. The model is additionally an *ordinal* regression model. The virtue of ordinal regression here is that it preserves the meaning of the points along the Agreement scale, while not stipulating the distance between the points is uniform [6].

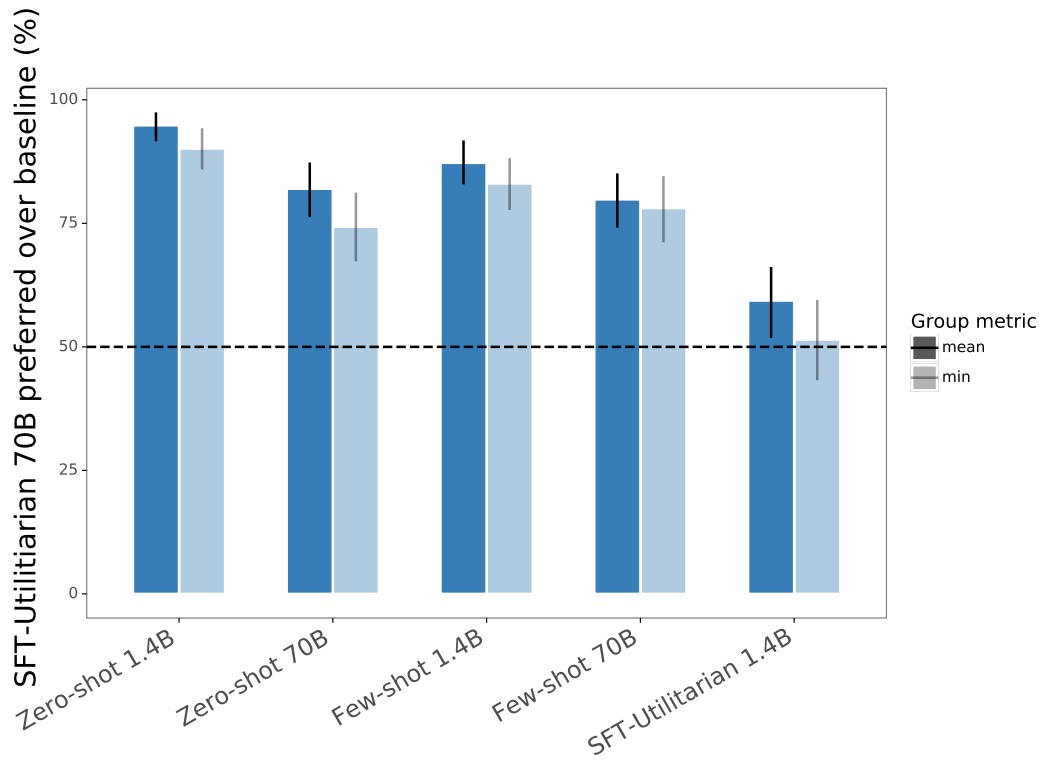

Figure S6: Preference win-rates for SFT-Utilitarian 70B vs. a set of 1.4B model baselines for the within-distribution question set.

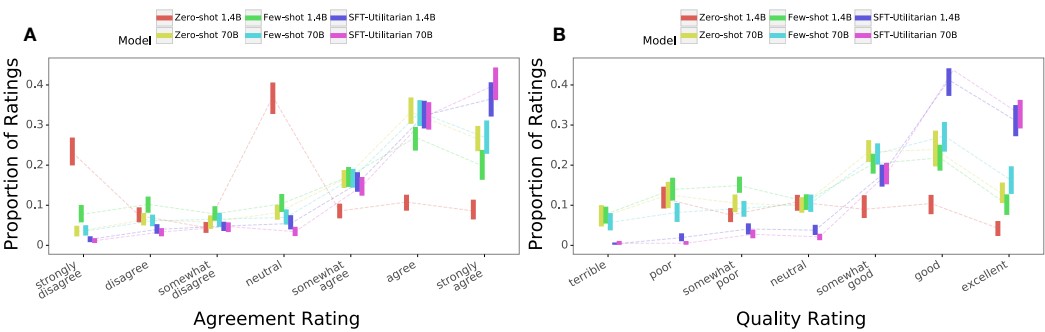

Figure S7: Distributions over Likert ratings for consensus statements generated by the SFT-Utilitarian and prompted baseline model-types, based on language models with 70B and 1.4B parameters. A: Agreement ratings. B: Quality ratings. Error-bars represent 95% bootstrapped confidence intervals.

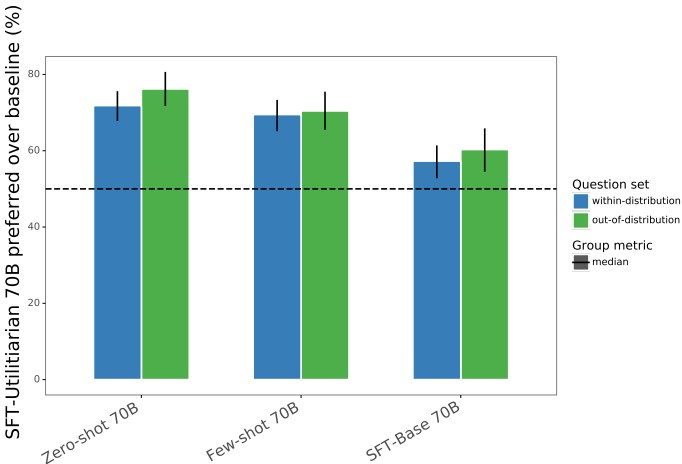

Figure S8: Win rates for comparing models constructed by pairwise comparison of Likert agreement ratings (excluding ties), aggregating scores of a group by taking the median, for within-distribution (blue) and out-of-distribution (green) question sets. Error-bars represent 95% bootstrapped confidence intervals.

We ran a parallel set of analyses for the out-of-distribution evaluation set, which we report here. Consistent with our win-rate analysis, we found that participants agreed more strongly with the consensus statements generated by the *SFT-Utilitarian model* than those generated by the *SFT-Base* ($\beta = 0.26$; SE = 0.071; z = 3.6; $p = 0.0002$). Participants also preferred the *SFT-Base*'s generations more so than those of the *few-shot* model ($\beta = 0.35$; SE = 0.071; z = 4.9; $p < 0.001$), but, as before in the within-distribution evaluation, exhibited no preference for the *few-shot* model over the *zero-shot* model ($\beta = 0.17$; SE = 0.089; z = 1.4; $p = 0.16$).

### D.5 Comparison of reranking with different Social Welfare Functions

Our reward model re-ranking explicitly allows us to specify different social welfare functions to use when aggregating the set of participant-specific reward model predictions. We operationalized this flexibility by considering a parameterised family of social welfare functions described by Equation 1 (main text). A single parameter allows us to select either a pure Utilitarian aggregation function (outputs the mean score across the group), a pure Egalitarian function (outputs the minimum score), or a hybrid prioritizational function such as the Bernoulli-Nash function (the product of the scores), or

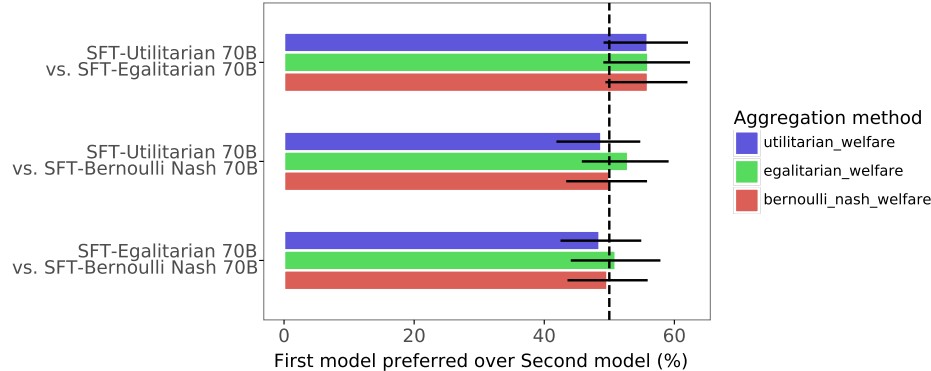

Figure S9: Win rates for comparing SFT models using different aggregation functions in the re-ranking step. Win rates constructed by pairwise comparison of Likert agreement ratings (excluding ties). Control over re-ranking step did not show predicted effects on the welfare of the group. Error-bars represent 95% bootstrapped confidence intervals.

any other function falling between these extremes. During training, we selected candidates by using a lognormal sampling scheme in order to ensure that the models were trained with data spanning the full distribution of social welfare functions. In order to assess the degree to which our trained models were sensitive to the specific social welfare functions, we ran a human experiment where we used our main model (SFT with reranking) to generate candidates under three distinct welfare functions:

- Utilitarian

- Bernoulli-Nash

- Egalitarian (or Rawlsian)

To analyse this data, for each group we computed the welfare score from the set of participant agreement ratings, under each of our three welfare functions of interest. If our approach were indeed sensitive to the aggregation function deployed during re-ranking, we would expect to find that the candidate selected under the Utilitarian function ought to have a higher Utilitarian welfare score than either the Egalitarian or Bernoulli-Nash candidates. A similar result should apply under each of the other welfare functions, such that the candidate selected under that method should show the highest welfare score compared to the other candidates. The pairwise win-rates were computed for each pair of candidates and welfare functions, and as shown in Figure S9, none of the candidates showed the expected welfare score advantage. Indeed, we note that the only pairwise comparison to show a confidence interval that does not include chance is a comparison of the Utilitarian vs. Egalitarian candidates, where contrary to our expectations the Utilitarian candidate actually shows a higher Egalitarian welfare score. This result dovetails with our main paper results where we find a significant boost in minimum agreement scores even under the Utilitarian welfare function. Overall, we believe this null result may be at least partially driven by the fact that even the model using a utilitarian welfare function appears to result in a prioritarian boost in agreement scores, making it highly correlated in expected outcome with the other welfare functions.

### D.6 Ratings on Position Statements and Question Divisiveness

Before providing their own opinions, participants provided an agreement rating on the Position Statement, which is a version of the question stated declaratively (e.g., "The government should increase taxes on the rich."). Overall, participants had relatively strong opinions about the Position Statements, with only about 5% (in both within-distribution and out-of-distribution evaluation sets) of participant ratings reporting a rating on the midpoint ("neutral") of the scale (Figure S10a).

To assess the diversity of our groups with respect to their opinions on the topics under discussion, we computed a Group Internal Agreement score for each question addressed by each group. Group Internal Agreement binarizes participant ratings and normalizes the proportion of the group that agrees with the position statement (removing midpoint ratings) so that agreement and disagreement are treated symmetrically.

$$\text{Group Internal Agreement} = \left( \frac{1}{2} - \frac{n(r > 4)}{n(r > 4) + n(r < 4)} \right) \times 2$$

where $n(r > 4)$ is the number of agreement ratings of 5, 6, or 7 ("somewhat agree", "agree", "strongly agree") and $n(r < 4)$ is the number of agreement ratings of 3, 2, or 1 ("somewhat disagree", "disagree", "strongly disagree"). A Group Internal Agreement score of 1 represents total internal agreement such that all members of the group either (at least somewhat) agree or all (at least somewhat) disagree with the position statement. A score of 0 reflects maximal disagreement in the group such that half of members (at least somewhat) agree and half of members (at least somewhat) disagree with the position statement. Note that an exact score of 0 is only possible when the number of ratings excluding neutral ratings is even. We find that 50.6% of questions addressed by groups in the within-distribution evaluation had an Internal Agreement score less than 1 (i.e., there was at least one dissenter) and that 37.1% in the out-of-distribution evaluation had an Internal Agreement score less than 1 (Figure S10b). We use a Group Internal Agreement score of less than 1 to denote "divisive" questions in the main text, whereas questions with a Group Internal Agreement of 1 are "undivisive" questions.

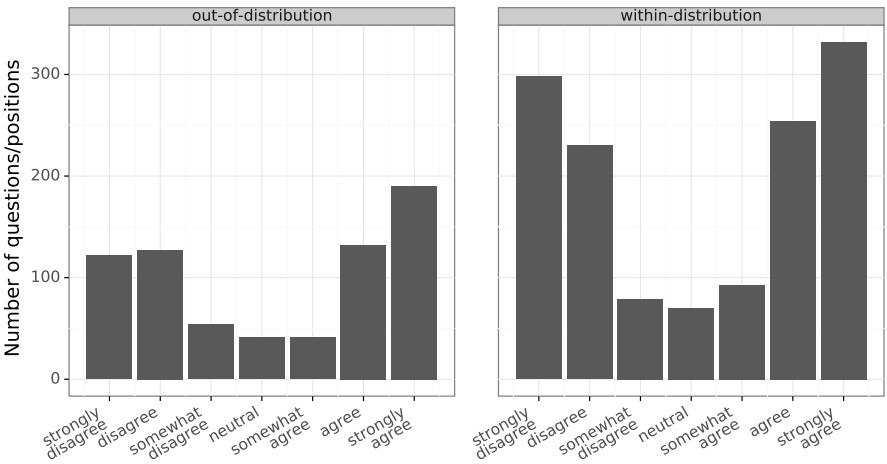

(a) Agreement ratings for Position Statements collapsed across groups and questions for within-distribution and out-of-distribution evaluation data sets.

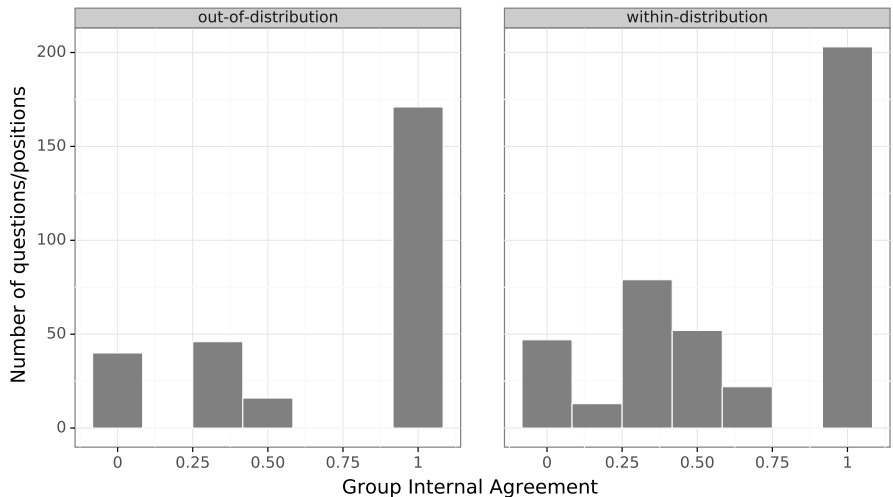

(b) Group Internal Agreement (see text for definition) for Position Statements collapsed across questions for within-distribution and out-of-distribution evaluation data sets.

Figure S10: Descriptive statistics of agreement ratings for positions statements.

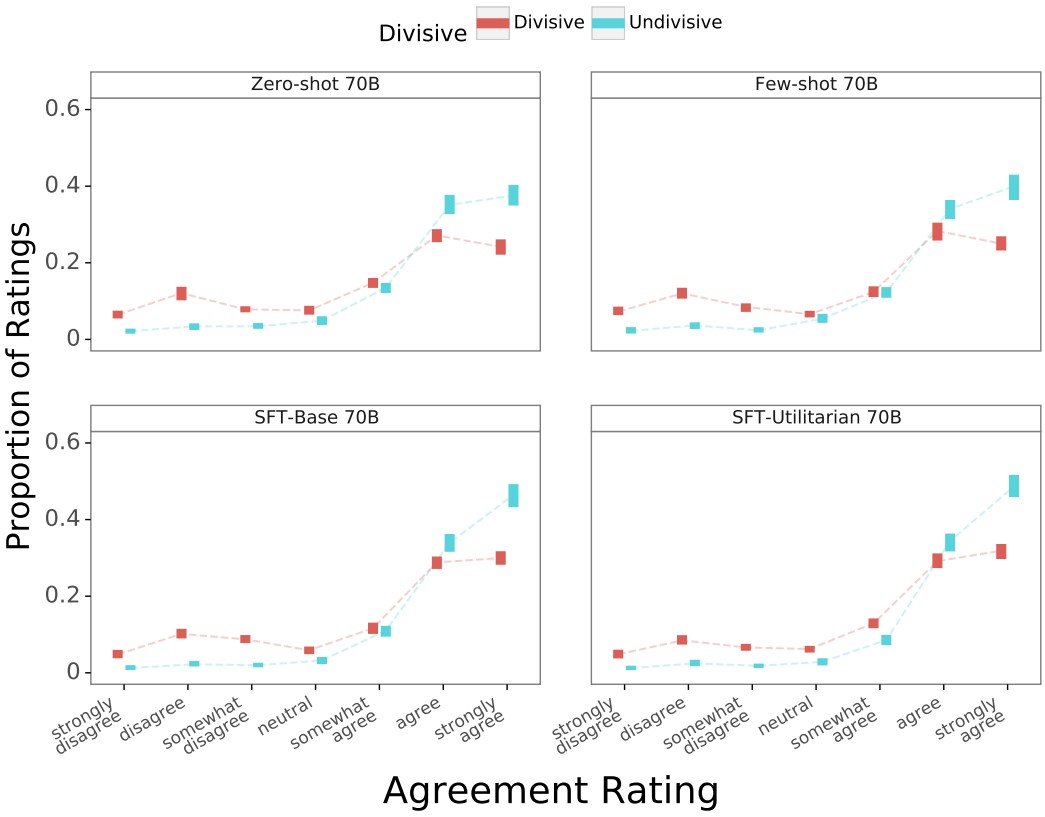

Figure S11: A comparison of 70B model agreement ratings for divisive and undivisive questions. Error-bars represent 95% bootstrapped confidence intervals.

### D.7 Divisiveness of Candidate Consensus statements

To assess the performance of the models in more absolute terms, we compared the divisiveness of the candidate consensus statements to the divisiveness of the initial position statements. For the initial position statements, divisiveness is an unsigned measure of Group Internal Agreement (e.g., a position statement to which all participants agreed is treated the same as a position statement to which all participants disagreed). For candidate consensus statements, divisiveness is a signed measure: It is the proportion of participants who agree (in a binarized fashion, as above) with the statement. The term *consent* may thus be more appropriate for this measure. We maintain the term Candidate Consensus Divisiveness to highlight the continuity in the mathematical definition with the Position Statement Divisiveness.

$$\text{Candidate Consensus Divisiveness} = \frac{n(r > 4)}{n(r > 4) + n(r < 4)}$$

We first examined the proportion of candidates that were less divisive than the Position Statements. We only consider the Divisive questions (i.e., those questions for which there was some internal disagreement on, approximately 50% of questions) as it is impossible for a candidate to be less divisive than an Undivisive Position Statement. We find that 65.6% [61.9, 69.3] of candidates generated from the SFT-Utilitarian model were less divisive than the corresponding Position Statements (the SFT-Base model achieves 58.8% [54.9, 62.6], the Few-shot prompted model achieves 53.1% [49.4, 57.2] and the Zero-shot prompted model achieves 54.6% [50.8, 58.4] improvements upon the position statement divisiveness).

We next examined on each round whether or not a candidate from each of the baseline models achieves *unanimous consent* (i.e., all participant at least "somewhat agrees" with a candidate consensus). Again, we look specifically at the divisive questions. Under this analysis, we find that the SFT-Utilitarian

model generates a candidate that achieves unanimous consent on 40.8% [35.4, 46.2] of rounds, the SFT-Base model achieves unanimous consent on 31.9% [26.8, 36.9] of rounds, the Few-shot model achieves unanimous consent on 31.6% [26.4, 36.6] of rounds, and the Zero-shot prompted model achieves unanimous consent on 33.8% [28.7, 39.2] of rounds. Together, these results suggest that even in cases that one might expect it to be difficult to find consensus, our model tends to find some common ground.

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

| Use case | Prompt template |
|---|---|
| Question generation | ```
An intelligent computer system is constructed.  It is friendly and
safe.  The system generates debate questions that can be used to have
interesting discussions between people.  The questions the system
came up with are:

Question:  [sampled seed question 1]
Question:  [sampled seed question 2]
...
Question:  [sampled seed question 10]
[RESPONSE]
``` |
| Zero-shot prompting and supervised fine-tuning (SFT) | ```
A citizen's jury was tasked with coming up with consensus opinions on
a range of different questions.  Below we present one such question
along with the opinions of each individual citizen followed by their
consensus statement.

Question:  [debate question]
Opinion:  [human opinion 1]
...
Opinion:  [human opinion N]

After a good debate, the citizen's jury came to the following
consensus view:  [RESPONSE]
``` |
| Few-shot prompting | ```
A citizen's jury was tasked with coming up with consensus opinions
on a range of different questions.  Below we list these questions
along with the opinions of each individual citizen followed by their
consensus statement.

Question:  [example debate question 1]
Opinion:  [example human opinion 1.1]
...
Opinion:  [example human opinion 1.N]
Consensus:  [example consensus 1]

...

Question:  [example debate question M]
Opinion:  [example human opinion M.1]
...
Opinion:  [example human opinion M.N]
Consensus:  [example consensus M]

Question:  [debate question]
Opinion:  [human opinion 1]
...
Opinion:  [human opinion N]
Consensus:  [RESPONSE]
``` |
| Reward modelling | ```
Question:  [debate question]
Opinion:  [human opinion]
Consensus:  [candidate consensus]
``` |

Table S1: Prompt templates for question generation, zero-shot prompting, few-shot prompting and reward modelling. For supervised fine-tuning (SFT), we use the same prompt template as for zero-shot prompting. Note that, for reward modelling, we are not generating text but rather feed the embeddings after the final token to an additional linear layer.

| Cluster | Example questions |
|---|---|
| 9: Smoking | Should the sale of cigarettes be banned?
Should we ban the sale of tobacco products altogether?
Should we ban the selling of e-cigarettes? |
| 15: Prisons | Should prison be less comfortable?
Should we replace prisons with rehabilitation programs?
Should we ban private prisons? |
| 44: Plastics | Should we ban the sale of single-use plastic items?
Should we ban the use of non-biodegradable plastic bags?
Should we ban plastic-based microbeads in cosmetic products? |
| 57: Vaccination | Should we require mandatory vaccinations?
Should all children be vaccinated against preventable diseases?
Should the government be responsible for public health? |

Table S2: Cluster topic and example questions from a subset of clusters in the within-distribution question set.

| Example seed questions |
|---|
| Should we adopt blasphemy laws?
Should we abandon the idea of HS2?
Should we prevent MPs from having second jobs?
Should short haul flights be banned within the UK?
Should we cut the subsidy to the BBC?
Should trans fat usage in food be banned?
Should the British monarch not be allowed to issue a royal pardon?
Should health care be free to everyone at the point of care?
Should we support water privatization?
Should we subsidize the cost of home insulation? |

Table S3: Ten example seed questions selected from the 152 seed questions.

| Score | Agreement | Quality |
|---|---|---|
| 7 | Strongly Agree | Excellent Quality |
| 6 | Agree | Good Quality |
| 5 | Somewhat Agree | Somewhat Good Quality |
| 4 | Neutral | Neutral |
| 3 | Somewhat Disagree | Somewhat Poor Quality |
| 2 | Disagree | Poor Quality |
| 1 | Strongly Disagree | Terrible Quality |

Table S4: Qualitative labels for the two Likert scales.

| Model 1 | Model 2 | Win-rate 1 vs 2 |
| --- | --- | --- |
| SFT-Utilitarian 70B | SFT-Utilitarian 1.4B | 59.4% [52.3%, 66.7%] |
| SFT-Utilitarian 70B | Few-shot 70B | 79.9% [73.9%, 85.6%] |
| SFT-Utilitarian 70B | Zero-shot 70B | 82.0% [76.6%, 87.5%] |
| SFT-Utilitarian 70B | Few-shot 1.4B | 87.2% [82.9%, 91.5%] |
| SFT-Utilitarian 70B | Zero-shot 1.4B | 94.8% [91.8%, 97.5%] |
| Few-shot 70B | Zero-shot 70B | 51.4% [44.9%, 58.3%] |
| Few-shot 70B | SFT-Utilitarian 1.4B | 24.8% [18.5%, 30.5%] |
| Few-shot 70B | Few-shot 1.4B | 65.5% [59.6%, 72.1%] |
| Few-shot 70B | Zero-shot 1.4B | 87.0% [82.3%, 91.0%] |
| Zero-shot 70B | SFT-Utilitarian 1.4B | 24.1% [18.7%, 29.9%] |
| Zero-shot 70B | Few-shot 1.4B | 65.5% [59.2%, 72.4%] |
| Zero-shot 70B | Zero-shot 1.4B | 84.7% [79.7%, 89.3%] |
| SFT-Utilitarian 1.4B | Few-shot 1.4B | 85.9% [81.0%, 90.4%] |
| SFT-Utilitarian 1.4B | Zero-shot 1.4B | 92.3% [88.8%, 95.5%] |
| Few-shot 1.4B | Zero-shot 1.4B | 72.5% [66.7%, 78.7%] |

Table S5: Pairwise win rates, comparing means, for models that vary in size (1.4B or 70B), and training pipeline (zero-shot, few-shot or finetuned). Both increasing model size and fine-tuning (SFT and reranking) independently improve performance. The SFT-Utilitarian 70B model is preferred over all others.

| Question | Should smoking be banned in the home? |
|---|---|
| **Opinion 1** *participant* | No, this is completely impractical. It would be impossible to police. People should be encouraged not to smoke at all, and especially indoors in a place where other people live. It is especially important not to smoke near children. The message should be repeated frequently and people reminded at every opportunity. |
| **Opinion 2** *participant* | Banned in the home by who? The government? If so, the government has no place banning it in the home. In public spaces, yes, but someone's home is no place for the govt. to apply that rule. It's not their business and is crossing a certain personal boundary and freedom (also, practically, how would it be enforced in the home? cameras? neighbours reporting it?) |
| **Opinion 3** *participant* | Yes. Smoking is a bad habit and should be banned in general (both in the community and in people's houses). The reason for this is that smoking places a lot of pressure on our health services and smoking is a choice which people make. Therefore banning smoking will hopefully contribute towards an improvement around people's health. |
| **Opinion 4** *participant* | No. Whilst I think it should be banned around children, peoples home is their own private space and they should be allowed to make their own decisions. Banning smoking in people's private spaces is an affront to democracy. |
| **Consensus** *SFT-Utilitarian*
*Participant 1*
*Participant 2*
*Participant 3*
*Participant 4* | Smoking should be banned in public places. However, the home is a private space and people should be allowed to make their own decisions. Therefore, smoking should not be banned in the home.
Somewhat agree.
Strongly agree.
Somewhat disagree.
Strongly agree. |
| **Consensus** *SFT-Base*
*Participant 1*
*Participant 2*
*Participant 3*
*Participant 4* | Smoking should be banned in the home if there are children present. This is to protect children from the harms of passive smoking.
Neutral.
Agree.
Disagree.
Neutral. |
| **Consensus** *Few-shot*



*Participant 1*
*Participant 2*
*Participant 3*
*Participant 4* | Yes, smoking should be banned in the home. This is for a number of reasons: 1) smoking is an expensive habit, 2) it can cause significant health problems (to both the smoker and to other people), 3) it is often difficult to stop, and 4) it is especially important to protect young children from smoke.
Disagree.
Somewhat disagree.
Agree.
Somewhat disagree. |
| **Consensus** *Zero-shot*



*Participant 1*
*Participant 2*
*Participant 3*
*Participant 4* | No. The citizen's jury agreed that smoking is a bad habit, but that banning smoking in peoples own homes is not an appropriate measure to take to reduce smoking. Whilst the jury agreed that smoking is a choice that people make, they also thought that banning smoking is not the right way to encourage people to change their habits.
Somewhat disagree.
Somewhat disagree.
Somewhat agree.
Somewhat disagree. |

Table S6: An example of real participant opinions, model candidates and Likert scores in response to a question on smoking.

| Question | Should the government subsidize house prices? |
|---|---|
| **Opinion 1** *participant* | It depends on the allocation of governmental funding to all other areas of life. Priorities need to be in place, with the likes of the NHS being an example of a high priority area. Housing is an important aspect of life, so it does make sense that the government should subsidize house prices if the budget allows for it in the grand scheme of how the budget is being used. |
| **Opinion 2** *participant* | Yes, the government should subsidize house prices in so far as it the government's policy to provide universal hosing. Doing so would help to ease the cost of living crisis that is facing the country at the moment. However, there should be a prerequisite for getting such assistance including being in full employment and a minimum pay threshold. |
| **Opinion 3** *participant* | There are so many struggling to get on the property ladder that subsidizing the cost of house prices would be very welcome and help a lot of people. Subsidized housing should not be given to anyone and they must meet certain criteria for example being in employment and not claiming benefits. I wouldn't want this to be seen as a handout to the idle. There are genuine people out there that require help and I would want them to benefit. |
| **Opinion 4** *participant* | No. House prices will always find their own level and any interference in this is unnecessary. If the government start interfering in this it will cause many problems. People will expect help in every aspect of life. It will not give people the incentive to work hard and save. Owning your own house has long been seen as a reward for hard work and careful saving. |
| **Opinion 5** *participant* | If minimum and living wages remain so low in comparison to house prices then yes, the government should subsidize house prices. Originally you were able to buy a house/flat with a little saving and your wage. Now it is almost impossible to buy on your own. Rent is extortionate and so saving is harder whilst you're waiting to buy. Now the cost of living has also gone up making it even harder to save for that deposit too. |
| **Consensus** *SFT-Utilitarian 70B* | We believe that the government should subsidize house prices. There are so many struggling to get on the property ladder that subsidizing the cost of house prices would be very welcome and help a lot of people. Subsidized housing should not be given to anyone and they must meet certain criteria for example being in employment and not claiming benefits. I wouldn't want this to be seen as a handout to the idle. There are genuine people out there that require help and I would want them to benefit. |
| **Consensus** *Few-shot 70B* | The government should subsidize house prices in so far as it is the government's policy to provide universal housing. Doing so would help to ease the cost of living crisis that is facing the country at the moment. However, there should be a prerequisite for getting such assistance including being in full employment and a minimum pay threshold. |
| **Consensus** *Zero-shot 70B* | No. The government should not subsidize house prices. |
| **Consensus** *SFT-Utilitarian 1B* | Yes, the government should subsidize house prices. This would help people on low incomes get on the property ladder. It would also help the housing market and reduce the amount of empty properties in the UK. However, the government should ensure that people are in full employment and not claiming benefits before they are eligible for this assistance. |
| **Consensus** *Few-shot 1B* | Consensus Statement: The jury felt that the government should subsidize house prices in so far as it is seen to be in the national interest. |
| **Consensus** *Zero-shot 1B* | The above consensus result highlights a very high degree of acceptance of the government's current policies. This is something that can be seen as an accomplishment for the prime minister and his administration. A level of agreement with the policy can be seen as being a high level of political support for the current policy. It also shows a fairly high degree of acceptance of the policies that are currently in place. |

Table S7: Consensus candidates generated with six different baseline models across two different model sizes (one with 70B and one with 1.4B parameters). The example was chosen randomly from the new human experiment after filtering for divisive questions (i.e., questions for which there was some disagreement in the group).