# OpenReview forum: "Fine-tuning language models to find agreement among humans with diverse preferences"
_NeurIPS.cc/2022/Conference — NeurIPS 2022 Accept_

### Official Review · Reviewer_bGmJ · 2022-07-08

**Rating:** 3
**Confidence:** 4
**Soundness:** 2 fair
**Presentation:** 3 good
**Contribution:** 2 fair

**Summary:**

Motivated by the fact that people could express diverse opinions on a topic, this work explores whether a LLM would be capable of generating consensus statements from such differing opinions. As part of their extensive data collection initiative, the authors collect opinions from participants on moral and political issues specific to the UK. Consensus statements are first generated using a Chinchilla model, with opinions as prompts. The generated statements are rated by the participants for agreement and quality, and high-quality statements are used to fine-tune a pretrained Chinchilla model. A reward model is then trained to rank consensus statements in order of their perceived appeal to the overall group of participants. The work demonstrates that the model-produced consensus statements significantly outperform those obtained by their baseline models.

**Questions:**

- It would be helpful if the authors could motivate and provide a short overview about the Social Welfare Functions used, e.g. Rawlsian, Utilitarian (lines 141-152). These would potentially be new concepts to ML practitioners and it would be helpful to briefly understand the motivation for using these specific functions.
- Lines 168-170: The authors mention omitting a few opinions to ensure data diversity. Could they please elaborate further on why they adopted this strategy, since this would reduce the number of diverse opinions the model would be allowed to see?
- The core of the study is based upon the fact that participants provide diverse opinions to the questions in the corpus. Could the authors please elaborate on how diversity was ensured? Is there a quantitative measure they could report for this?

**Limitations:**

The Limitations section has been well-written.

**Strengths And Weaknesses:**

Strengths:
- The paper is well-written and easy to comprehend.  The underlying problem is well-explained and the methodology has been clearly presented.
- It is great to observe that ethical considerations were taken into account during the data collection efforts (lines 128-133).
- The authors are aware of the potential biases that may arise and have detailed them in the Limitations section. For instance, lines 314-315 mention the likelihood of users with authoritative tones commanding more influence over the consensus. Further, the limitations w.r.t. the specifics of the data study cohort are well etched out in lines 325-333.
- Potential risks of this work have also been touched upon in lines 342-348, where the authors mention the potential for misusing such LMs for persuasion.

Weaknesses:
Although well-written, the work unfortunately does not showcase any significant advancements in ML/NLP. It amounts to fine-tuning (a very specific) LLM for a specific task to show improvements over baselines. Questions arise on the choice of the LM made and the modeling strategy, along with the evaluation criteria used.
- Instead of using machine-generated consensus statements and further obtaining human annotations on them for finetuning the LM (lines 164-183), a more obvious alternative would have been to obtain ground-truth consensus statements from human participants and fine-tuning the LM on them. This set of participants could come from a different pool to the cohort who provided the differing opinions (i.e. the input). This would allow for the measurement of usual metrics for generation tasks, such as ROUGE, BLEU, METEOR etc. It is unclear why this was not selected as the modeling strategy for the paper.
- The motivation for selecting Chinchilla as the only LLM to experiment on needs some explanation. Have the authors explored other generative models such as BART, T5, GPT-2 etc for this task? Performance on these models should at least be reported as baselines.
- The evaluation criteria currently involves participants rating their preference of the consensus statements generated by the model in comparison with the baselines. This enforces the assumption that the statements generated by the model are satisfactory, both in terms of factuality and coherence. A stronger quality check on the generated text would be to compare these against human-generated consensus statements, and report ROUGE, BLEU/METEOR scores.

---

> ### Author Response · Authors · 2022-08-02
> **Response to Reviewer bGmJ (1/3)**
>
> We thank the reviewer for their thoughtful comments. We are glad they found the paper and methodology straightforward to digest, and we appreciate that they found the ethical considerations sufficiently thorough.
>
> Here we hope to clarify our choices regarding the LM, the modeling strategy, and the evaluation criteria.
>
> **“Instead of using machine-generated consensus statements and further obtaining human annotations on them for finetuning the LM (lines 164-183), a more obvious alternative would have been to obtain ground-truth consensus statements from human participants and fine-tuning the LM on them. … This would allow for the measurement of usual metrics for generation tasks, such as ROUGE, BLEU, METEOR etc.”**
>
> The modeling approach that the reviewer suggests, supervised fine-tuning with human-generated (“ground truth”) consensus statements, could have been an interesting alternative strategy to explore. However, we believe that our ‘learning from human feedback’ approach is more appropriate for our task for two main reasons.
>
> 1. Our goal is to build a system that generates high-quality statements that maximise welfare (group-level agreement). The ground truth metric for success under this objective is the agreement of the individuals who provided their opinions. Learning from human demonstrations could thus create misalignment between the fine-tuning objective (maximising likelihood of a demonstration, i.e., a human-generated consensus) and the true objective (maximising agreement). Previous results from the summarization literature (the most similar application to consensus generation) suggest this is likely to be the case. For example, Stiennon et al. 2020 found that models learning from human feedback outperform those learning from human demonstrations alone. This finding holds in other contexts beyond summarization, where human feedback also yields a robust improvement (Ouyang et al. 2022, Nakano et al 2022, Menick et al. 2022). One helpful insight to see why human feedback is more appropriate is that only learning from demonstrations incentivizes models to place probability mass on all demonstrations, including low-quality and low-agreement ones.
>
> 2. In addition to performance, human consensus writers will likely introduce bias in two important ways: (A) human consensus writers will each differ in how they aggregate individual opinions into a single consensus statement (i.e., what implicit Social Welfare Function [SWF] they use; e.g., focussing on aligning with the majority opinion or, alternatively, trying to include parts of the opinions of all participants); one could give explicit instructions to human consensus writers about how to do this weighting, but we have the additional goal of generalising across different SWFs and controlling the behaviour of the system via the SWF, and it is nonobvious how to use human demonstrations to achieve this goal .  (B) human consensus writers will have their own opinion on each question which (perhaps unconsciously) will influence the consensus statements in subtle ways.
>
> Nonetheless, we believe the reviewer’s suggested fine-tuning strategy with human demonstrations could be useful 1) to create an initial model that could be used for further fine-tuning using human feedback. We now use few-shot prompting as the initial model but for smaller models without strong few-shot capabilities, learning from human demonstrations will likely work better. 2) as a baseline training strategy to see if, similar to other tasks, learning from human feedback does indeed outperform a training strategy in which you only learn from demonstrations. We will include a discussion of this point in the camera-ready.
>
> That being said, the prospect of using human-generated consensus statements raises difficult data collection questions, since there are no available large external datasets for this task. We have done some initial tests, but this is a difficult task that will likely require specialised writers and large changes to our experimental design. It is certainly an interesting domain for future work.

---

> ### Author Response · Authors · 2022-08-02
> **Response to Reviewer bGmJ (2/3)**
>
> **“The motivation for selecting Chinchilla as the only LLM to experiment on needs some explanation. Have the authors explored other generative models such as BART, T5, GPT-2 etc for this task? Performance on these models should at least be reported as baselines.”**
>
> This is a great question and relates to a question asked by Reviewer 3uTu. To clarify our choice of model: We selected a large model similar to GPT-3 for its zero-/few-shot prompting capabilities. This capacity allowed us to bootstrap in the first iteration of our training pipeline without needing any human demonstrations (see paragraph 3.4 in the main text). We agree with the reviewer that we should have clarified our model choice better and we will add clarification in the camera ready version.
>
> We agree that it is still useful to include a smaller model as a baseline so we conducted an additional experiment in which we compare the 70B model head-to-head with a 1.4B version of our model, equivalent in size to GPT-2 but with the same architecture and dataset as our 70B model for a more controlled scale comparison  (see Hoffman et al. 2022). We compared consensus candidates generated by 6 models: (i) SFT-Utilitarian 70B (ii) Few-shot 70B (iii) Zero-shot 70B (iv) SFT-Utilitarian 1.4B (v) Few-shot 1.4B and (vi) Zero-shot 1.4B. Here, SFT-Utilitarian 70B corresponds to our best model, the fine-tuned model with welfare-based reranking (see paper for details). For the 1.4B parameter models,  both the generative model and the reward model are scaled down to 1.4B parameters.
>
> We have included the full figures in a new “Rebuttal.pdf” appendix but we will summarise our findings here. SFT-Utilitarian 70B still significantly outperforms each baseline in win rate when comparing mean agreement. Interestingly, SFT-Utilitarian 1.4B performs only slightly worse (the 70B version wins over the 1.4B version 59.4% [53.7%, 64.7%] of the time). This effect size is similar to SFT-Utilitarian 70B versus SFT-Base 70B (reported in the original submission), suggesting that both scale and reward modelling are independently beneficial for consensus generation. Additionally, SFT-Utilitarian 1.4B outperforms both zero-shot and few-shot 70B (win rates of 75.9% and 75.2% respectively). We will add these results to the camera ready version.
>
> Note, finally, that we did not rerun the full training and data collection pipeline for 1.4B (i.e. we did not first collect human data with 1.4B zero-shot and 1.4B few-shot prompting data and then two rounds of 1.4B model fine-tuning) but instead fine-tuned with data that we collected using the 70B model. Given that the smaller model has significantly weaker prompting capabilities, running the full pipeline with this small model would have probably made the process much more data intensive, as we could not have bootstrapped the high prompted performance of the 70B model.
>
> **“The evaluation criteria currently involves participants rating their preference of the consensus statements generated by the model in comparison with the baselines. This enforces the assumption that the statements generated by the model are satisfactory, both in terms of factuality and coherence. A stronger quality check on the generated text would be to compare these against human-generated consensus statements, and report ROUGE, BLEU/METEOR scores.”**
>
> As we expressed above, collecting human-generated consensus statements is a worthwhile enterprise, but one that is prohibitively complex to address in this submission. Metrics for evaluating the generated output from language models like ROUGE, BLUE, METEOR, while useful for many applications, have received criticism for poor correlation with human judgements in the context of summarization, the task closest to consensus generation (N. Schluter. 2017, Paulus et al. 2017, Chaganty et al. 2018, Stiennon et al 2020). For example, Figure 7 of Stiennon et al. 2020 shows how ROUGE fails to track summary quality as the models improve. In our task, where the goal is to generate high-quality consensus statements that maximise group-level agreement, metrics based on n-gram overlap like ROUGE and BLUE miss important features such as sensitivity to the polarity of a sentence (e.g., “We should lower the voting age.” would receive a high score if the reference statement is “We should not lower the voting age.”, even though participants that agree with one would disagree with the other). Because we are interested in maximising group-level agreement amongst participants, we believe that asking participants directly whether they agree is a more appropriate measure than using a ROUGE or BLEU with respect to a human-written reference statement. Additionally, we note that in addition to “agreement” we also collect “quality” ratings which are closer to “coherence” as the reviewer suggests.
>
> _[continued on next post]_

---

> ### Author Response · Authors · 2022-08-02
> **Response to Reviewer bGmJ (3/3)**
>
> _[continued from previous post]_
>
> However, we do agree with the reviewer that there are useful applications of ROUGE and BLEU scores which could help us understand our data better. We add a baseline model where we use ROUGE/BLEU between a participant’s opinion and a generated statement to predict participants’ agreement ratings (under the assumption that people will like statements that borrow more heavily from the text of their own opinion). Here, ROUGE/BLEU scores have a pairwise accuracy for choosing the highest rated consensus candidate of ~60% compared to ~72% for the RM. This comparison shows that ROUGE/BLEU is correlated with agreement but fails to capture it completely, consistent with the observations of Stienennon et al. (2022). (Note that we do not include METEOR as it is order-sensitive (designed for machine translation) while ordering of arguments should not matter for consensus.)
>
> **“It would be helpful if the authors could motivate and provide a short overview about the Social Welfare Functions used, e.g. Rawlsian, Utilitarian (lines 141-152).”**
>
> We absolutely agree a better explanation of these terms and concepts would be beneficial given the diverse readership of NeurIPS. We will add a more comprehensive explanation to the camera-ready version, explaining that cardinal social welfare functions are functions that take numeric representations of individual utilities as inputs and return a numeric representation of collective welfare, which we borrow from the field of economics. We will then address and motivate Utilitarian (Benthamite or max-sum), Egalitarian (Rawlsian or max-min) and Bernoulli-Nash (max-product) aggregation methods to provide additional context for readers.
>
> **“Lines 168-170: The authors mention omitting a few opinions to ensure data diversity. Could they please elaborate further on why they adopted this strategy, since this would reduce the number of diverse opinions the model would be allowed to see?”**
>
> Here by ‘diversity in the data’ we were specifically referring to diversity in the number of opinions the model was exposed to, rather than diversity in the content of those opinions. This was a deliberate design choice to ensure that the model would see a variable number of opinions during fine-tuning.
> We agree that this passage was not clear, and will alter it to the following:
> “To ensure that our dataset contains data on a variable number of opinions (between 3 and 5), each time a statement was generated we silently omitted 0, 1, or 2 of the participants’ opinions.”
>
> **“The core of the study is based upon the fact that participants provide diverse opinions to the questions in the corpus. Could the authors please elaborate on how diversity was ensured? Is there a quantitative measure they could report for this?”**
>
> We chose our domain of studying UK political questions because we expected our UK participants would hold diverse viewpoints about these questions. Thus, diversity was not ensured, but it was expected and confirmed through our measures of participants’ initial ratings about the Position Statements (Section 4.1). In the appendix (Section D.4), we described our measure of diversity (Group Internal Agreement), which measures how divisive the question is for a particular group of participants (which depends upon the specific 3-to-5 participants in a group), as opposed to a population-level measure of question divisiveness. We report that approximately 50% of our {group, question} pairs have a non-trivial level of diversity in opinion. We also note that our measure of diversity is a rather conservative measure of disagreement, as it only looks at the sign of participants' Position Statement agreements (i.e., agree vs. disagree), as opposed to the sign and magnitude (e.g., somewhat agree vs. strongly agree).
>
> **References**
>
> A. T. Chaganty et al. The price of debiasing automatic metrics in natural language evaluation. arXiv preprint arXiv:1807.02202, 2018.
>
> J. Hoffmann et al. Training compute-optimal large language models. arXiv preprint arXiv:2203.15556, 2022.
>
> J. Menick et al. Teaching language models to support answers with verified quotes. arXiv preprint arXiv:2203.11147, 2022.
>
> R. Nakano et al. WebGPT: Browser-assisted question-answering with human feedback. arXiv preprint arXiv:2112.09332, 2021.
>
> L. Ouyang et al. Training language models to follow instructions with human feedback. arXiv preprint arXiv:2203.02155, 2022.
>
> R. Paulus, et al. A deep reinforced model for abstractive summarization. arXiv preprint arXiv:1705.04304, 2017.
>
> N. Schluter. The limits of automatic summarisation according to rouge. In Proceedings of the
> 15th Conference of the European Chapter of the Association for Computational Linguistics, 2017.
>
> N. Stiennon et al. Learning to summarize with human feedback. Advances in Neural Information Processing Systems, 2020.

---

### Official Review · Reviewer_HNUh · 2022-07-11

**Rating:** 6
**Confidence:** 4
**Soundness:** 3 good
**Presentation:** 2 fair
**Contribution:** 4 excellent

**Summary:**

The authors present an experiment for adapting a pre-trained language model (“Chinchilla”) to generate a consensus opinion given a set of individual opinions on a target question. Their method for generating consensus statements relies on 3 core components:
a) Use Chinchilla to generate candidate consensus statements. This model is (optionally) conditioned on the opinions of all/some individuals.
b) Use a reward model (RM; also based on Chinchilla) to predict the agreement score of each user (using the opinion of that user for conditioning) with each candidate statement.
c) Aggregate agreement scores using different social welfare functions (SWFs) and generate one final ranking over all candidate statements. Then they select the statement with the highest ranking. For training, they sample different functions from a parametric family of SWFs.

To train the models, the authors collect a novel dataset in two rounds of human-in-the-loop training iterations, where they use study participants give their opinions on a set of curated political questions (targeted to a UK audience), generate consensus statements online, and have participants rate those consensus statement based on their agreement with them and quality of writing. This protocol creates a number of training instances (comprising questions, individual opinions, consensus statements, agreement/quality ratings) that they then use to fine-tune their models before the next round.

A big part of their work is centered on evaluating their proposed approach.. They first look at average agreement and quality scores obtained for their proposed method and the baselines (proposed method is better), as well as the “winning ratio” (percentage of times where proposed method was preferred over each of the baselines; generally, higher than 50%). Additionally, they investigate the gains of finetuning, as well as the additive impact of the RM, and conclude that both are useful (with finetuning yielding the largest gains). Furthermore, they look into the impact of user consensus (undivisive questions, where users unanimously agree, and divisive ones, where there’s various degrees of disagreement) but find no major effect. They test whether the consensus opinions generated by their model are preferred by opinions (not consensus ones - see weaknesses) given by human annotators, and find that their model is preferred in the majority of times. Their last evaluation concerns the impact of excluding some opinions when generating the consensus (as a sanity check that their model is indeed utilizing all opinions).

Finally, they discuss limitations and opportunities of their proposed approach.

**Questions:**

1. The authors state (l. 122-123 in main text; Appendix B.2.1) that they had each participant rate each consensus prediction twice to measure intra-rater reliability (and subsequently discard raters with a low reliability). What is the time interval between those two ratings, and how many other instances did they rate in-between? In short, is it reasonable to assume that the two ratings were somewhat independent (i.e. the raters had “forgotten” their first rating by the time they did the
second?)?
2. l. 171 “each of which is ranked top-1 by a different $\alpha$”: does this ranking include the RM model to estimate the agreement of each rater?
3. Did you obtain new opinions and ratings from new participants for 4.2? This means you had an initial set of participants which generated the initial 4/5 opinions, from which the model created a consensus, and then 4 more participants who gave a second set of opinions (for comparison) and preference ratings? I think the authors are referring to the original participants but it’s not 100% clear from their formulation.
4. Can the authors provide estimates of agreement for divisive vs undivisive questions? Decisive questions should have, in general, much higher agreement scores, as it is trivial to produce a consensus (i.e. a simple yes/no without any further justification should be enough to get the participants to agree, since they all agreed/disagreed with the positive statement anyway).

**Limitations:**

While the authors do a very good job at covering potential limitations/biases of their work, there is one potential risk of misuse that they do not cover: It is possible to (mis)use their model to wrongly present a summary as the consensus opinion of a particular demographics. For example, one could aggregate several social media posts of that demographic, and generate a statement “Demographic X thinks the government should do Y”. This would be something other than “Persuasiona” and “Factuality”. A potential title for this type of abuse would be “Misrepresentation”. This is particularly pertinent as the authors removed questions that could potentially illicit offensive answers (for good reason), which means they are unable to test how their model functions in those cases.

**Strengths And Weaknesses:**

# Strengths

The paper’s main strength lies in its motivation (fine-tuning a language model to generate consensus opinions is a novel and interesting task) and the sheer computational and financial cost expended for this experiment (dozens of GPUs; 43k pounds spent for compensating human participants) making it an interesting contribution to the field of NLP as it investigates a very interesting application. The authors also deserve credits for highlighting a lot of the ethical issues that come with this application in Section 5 (What is consensus anyway? What are potential biases introduced by their recruiting protocol? Etc.). This casts this contribution as a work of significance in its field in terms of novelty of applications, though not in terms of theoretical advancement (which is fine, as it was not the goal here).

In addition, there are several strengths in how the authors conducted their experiments:

1. They made efforts to pay human participants a living wage (approx. 15 pounds an hour), which is appreciated in an era where similar works choose a pay-per-click model. This may potentially increase the quality of their data.
2. They held out an out-of-domain validation set for testing the generalization of their models to new topics.
3. They utilize a well-validated concept from economics for parametrizing consensus (the SWFs, which allow for different interpretations of consensus depending on the value of parameter alpha). This makes their method suitable for different formulations of consensus that might be needed for different applications.
4. They include an ablation analysis to see how different components improve performance (few-shot vs zero-shot, fine-tuning, reward model) and additionally test the differences with mixed-effects models.
5. They test various aspects of their models (see summary)

# Weaknesses

There is one main weaknesses that stand out in this work: The experiments are not reproducible, as a) the authors do not release their data/models/code and b) the cost of the experiment is prohibitive for most groups that would want to replicate this study. This does not diminish the importance of the contribution (as it is good that someone did it) but makes it harder to verify the main findings of this work in the future (Are language models capable of performing this task? What can we do to improve consensus? How else can we formulate consensus? What bias did this model introduce? Etc.).

Other weak points are:

1. Most of their analyses are done by comparing means, except Sec. 4.3, where the use of medians pops up all of a sudden (l. 279). In general, the use of the median as basis for comparisons should be preferred throughout as the distributions are not expected to be Guassian (e.g. in the case of divisive questions where the model optimizes for the Rawlsian SWF there should be a sort of bimodal distributions, with some raters agreeing and some disagreeing with the consensus)
2.. The comparison in 4.2 seems unfair. Authors compare model-generated consensus statements (which are learned to be more balanced) with participant opinions which often are, by the authors’ own motivation, biased. Moreover, in the case of undivisive questions, where participants should all more-or-less agree with one another, we should not expect to see a big difference (certainly not ~80% preference for model candidates).

## Presentation issues

Some minor presentation issues that can be easily improved:

1. Results section (l. 217-229) contains a lot of new information on the data collection process that should be moved further up (potentially in Sec. 3.2). In particular, the number of raters for each question should be mentioned before discussing how many opinions were presented to the model (l. 168)

---

> ### Author Response · Authors · 2022-08-02
> **Response to Reviewer HNUh (1/3)**
>
> We thank the reviewer for their thoughtful and positive review. We respond to the weaknesses and questions below point-by-point.
>
> **“The experiments are not reproducible, as a) the authors do not release their data/models/code and b) the cost of the experiment is prohibitive for most groups that would want to replicate this study”**
>
> We are sympathetic to the concerns of reproducibility and have tried our best to document all of the necessary steps to reproduce our results, given access to a large language model (LLM) and a sufficient computational budget and funds for human data collection. Although LLMs currently have limited availability, they are likely to become more accessible in the near future. In fact, in the time since the submission of this paper, the BigScience initiative released a public 176B parameter LLM called BLOOM (https://bigscience.huggingface.co/blog/bloom). Thus, insights from this study are likely to become increasingly relevant to the wider community over time.
> Further, we have now trained a smaller, GPT2-sized model (1.4B parameters) on this task, which shows good performance, outcompeting prompted versions of our larger 70B parameter model (please see our response to reviewer 3uTu for more details). While it is still not as effective as the larger trained model, it nevertheless performs well enough to allow some degree of replicability on our task - a job made substantially more feasible by the model’s smaller size.
>
> **“Most of their analyses are done by comparing means, except Sec. 4.3, where the use of medians pops up all of a sudden (l. 279). In general, the use of the median as basis for comparisons should be preferred throughout as the distributions are not expected to be Gaussian."**
>
> We agree with the reviewer that the distributions of Likert judgments are not expected to be Gaussian, and in general, the median would be a better metric for central tendency for our small groups of agreement scores. However, for these human evaluations, the model is explicitly trying to optimise for the mean (Utilitarian) welfare, so for ease of interpretation, we decided to keep with the mean as our primary measure of central tendency. Still, we agree that the median would be useful to report, and we include these additional analyses in the new “Rebuttal.pdf” appendix found in the updated supplementary material .zip file (Figure R1). Usage of the median results in the same substantive conclusions for the primary analyses.
>
> **“The comparison in 4.2 seems unfair. Authors compare model-generated consensus statements (which are learned to be more balanced) with participant opinions which often are, by the authors’ own motivation, biased. Moreover, in the case of undivisive questions, where participants should all more-or-less agree with one another, we should not expect to see a big difference (certainly not ~80% preference for model candidates).”**
>
> We acknowledge that this baseline is not perfect. We were explicit about both the potential weakness of this baseline and our justification for including it in our original submission at the start of section 4.2:
> “Here we compare the performance of our SFT-Utilitarian model against human-generated opinions. While the human opinions are not explicitly written with consensus in mind, the opinions are very high-quality, frequently containing well-reasoned justifications for their positions.”
> We agree with the fact that the opinions, by their nature, cover a range of different viewpoints and that makes the comparison of mean agreement with opinions vs. model candidates less useful. This led us to also compare each participant’s most preferred individual opinion vs. their most preferred model candidate. For example, given a particularly polarising topic, one would expect that an individual would select an opinion most similar to their own and that they could prefer this over an explicitly consensus-seeking model candidate. For this comparison, we find that our best model candidates were still preferred over the most preferred opinions 65% of the time.
> The reviewer is correct to point out that we should not expect to see a big difference for preference of model candidates over human opinions for undivisive questions. We looked at the divisiveness split for this evaluation, and found that the preference for most preferred candidates to most preferred opinions was 63.3% (57.4%, 68.9%), which was numerically smaller than for the divisive questions 66.2% (61.1%, 71.4%)... not the 80% figure the reviewer quoted (80% was the upper bound of the CI for the comparison of means). We will add this divisiveness split results to the camera-ready version for further clarification.

---

> ### Author Response · Authors · 2022-08-02
> **Response to Reviewer HNUh (2/3)**
>
> **“they had each participant rate each consensus prediction twice to measure intra-rater reliability (and subsequently discard raters with a low reliability). What is the time interval between those two ratings, and how many other instances did they rate in-between? In short, is it reasonable to assume that the two ratings were somewhat independent..?”**
>
> Participants first rate each candidate sequentially and then rate each candidate again (rate 6, then rate 6 again; the order within each set of 6 is randomised). Thus, the time between ratings is only a function of how long it takes participants to read and rate approximately 6 other candidates. At most, this delay is a few minutes long.
> Our intention here was to filter out ratings from participants that often varied hugely between their first and second ratings, as we consider this to be a proxy for the participant paying attention to the task (rather than responding randomly). We expected participants' two responses to be roughly the same, with some tolerance for noise, which informed our usage of a relatively lenient intra-rater reliability threshold. We will clarify this point in the camera ready version.
>
> **“L. 171 “each of which is ranked top-1 by a different α”: does this ranking include the RM model to estimate the agreement of each rater?”**
>
> This is correct. The first step of ranking involves using a reward model to output a score estimating how much a given player is predicted to agree with a given consensus statement. These reward scores are then aggregated with a social-welfare function (SWF, parameterised by alpha). Therefore, for each of the 16 candidates we generate, we compute the SWF score and then rank the candidates according to their score. In practice, to introduce SWF-diversity into our training pipeline, we sample two different alpha values, and rank the same 16 candidates twice, once under each value of alpha. We then pick the top-scored candidate under each alpha to present back to participants. We will add this clarification to the camera-ready.
>
> **“Did you obtain new opinions and ratings from new participants for 4.2? This means you had an initial set of participants which generated the initial 4/5 opinions, from which the model created a consensus, and then 4 more participants who gave a second set of opinions (for comparison) and preference ratings?”**
>
> For this evaluation, on each question each participant in the group of 5 wrote one opinion, and these (5) opinions were used to generate consensus statements. The same participants were then shown these model-generated consensus statements alongside the (anonymous) opinions of the other participants (4) in their group in response to that question. All participants then rate both the consensus statements and the opinions they see.
> This enables us to assess the extent to which a group of participants agreed with the consensus statement that explicitly took their opinion into account, versus human-generated opinions of other people responding to the same question. We will clarify this in the text to highlight that the opinions participants rated were from the same group in a round-robin style.
>
> **“Can the authors provide estimates of agreement for divisive vs undivisive questions? Decisive questions should have, in general, much higher agreement scores, as it is trivial to produce a consensus (i.e. a simple yes/no without any further justification should be enough to get the participants to agree, since they all agreed/disagreed with the positive statement anyway)."**
>
> In our new Figure R2 we now show that undivisive questions do indeed lead to higher agreement in the resulting consensus statement, across all models, consistent with the reviewer’s intuition. One subtle point, however, is that participants may broadly agree with the general stance (yes/no) of a statement, but disagree with the justifications or nuances given in the consensus. In these situations we expect there may be cases of unanimous agreement with the position statement, but disagreement with the resulting consensus statements, which tend to be more detailed. We avoid producing short and generic yes/no consensus statements by finetuning on statements that participants view as higher ‘quality’ (clear, coherent, self-justifying).

---

> ### Author Response · Authors · 2022-08-02
> **Response to Reviewer HNUh (3/3)**
>
> **"It is possible to (mis)use their model to wrongly present a summary as the consensus opinion of a particular demographics. For example, one could aggregate several social media posts of that demographic, and generate a statement “Demographic X thinks the government should do Y”."**
>
> We agree there is a risk that a user could misrepresent what the model is doing and assert that there is consensus among a particular demographic on a topic when in fact there is still significant disagreement. One way this could arise is if a user decided not to collect agreement ratings (as this data could in theory show if there was latent disagreement among the cohort of participants). The reviewer also refers to the risk that a human user misrepresents the consensus statements generated by the model by asserting that they speak for an entire demographic, rather than just the specific set of individuals the model directly engaged with. As the model is designed to reflect the views of those directly interacting with it, this case would also amount to a disingenuous presentation of the model. We agree that both of these potential misuses carry significant risks and we will add this limitation to the camera-ready version.

---

### Official Review · Reviewer_3uTu · 2022-07-12

**Rating:** 6
**Confidence:** 4
**Soundness:** 3 good
**Presentation:** 4 excellent
**Contribution:** 3 good

**Summary:**

This paper proposes to use a large language model (LLM) to find consensus between user statements; the main difference with previous summarization or opinion aggregation work is that the summary is evaluated by the users (people who generated the statements) themselves; thus the goal is to find common aspects in the statements that most users agree with.
The paper conducts a large-scale data collection with thousands of questions (political issues); human participants write statements about the questions and also evaluate the quality and agreement of candidate statements. Method-wise, the paper adopts a reranking model to assign rewards to consensus and uses it to rerank 16 consensus statements generated by another model. The consensus model is trained to generate high-quality summaries. The rewards model is trained to prefer statements with a higher agreement for a particular user (given the user's opinion). Results show that the model can beat zero-shot and few-shot baselines, as well as the model without the reranking step, for both quality and agreement ratings.

**Questions:**

1. The paper uses a quite large language model (70B parameters) for finetuning. Is this necessary and can we use smaller models? Usually, small models can also handle summarization and quality evaluation well if there is enough data, and smaller models are easier to deploy.

2. There is one step I do not quite understand about model training: in the first round, how is the top-1 candidate chosen from the 16 candidates? In the first round, we do not have the reward model yet, so it seems we do not have the utilities $u_1, …, u_n$ for ranking.

**Limitations:**

The authors state the limitation of the current work well and I agree with it.

**Strengths And Weaknesses:**

The paper presents an interesting idea of generating consensus, and collected high-quality data about political issues. The results clearly show gains over the baselines as well as human statements. The paper is well written and easy to follow.

The weakness is that the paper ignores many nuances about consensus generation and focuses solely on consensus generation. For example, one important question is the [detection] of consensus; is a consensus possible given the current statements or not? It is natural that for some questions and users, one cannot find much consensus for them to agree on. It is also interesting to verify the coverage of generated consensus (i.e., whether it covers all the points that the users agree on).

---

> ### Author Response · Authors · 2022-08-02
> **Response to Reviewer 3uTu (1/2)**
>
> We thank the reviewer for their thoughtful and positive review. We respond to the weaknesses and questions below point-by-point.
>
> **“The weakness is that the paper ignores many nuances about consensus generation and focuses solely on consensus generation. For example, one important question is the [detection] of consensus; is a consensus possible given the current statements or not? It is natural that for some questions and users, one cannot find much consensus for them to agree on.”**
>
> We appreciate the reviewer bringing up nuances about the phenomenon of consensus. We use the term ‘consensus’ to mean a statement that maximises group welfare under a given social welfare function. The term is often used in group decision-making settings (such as those discussed by Rawls or Habermas), however, to indicate a statement or proposal to which the group agrees, in a binary sense. Our setting is not set up to test for binary consensus (i.e., whether or not a group has “reached consensus”), but our model’s capacity to increase the minimum participant agreement levels (Fig. 2, main paper) suggest a strong ability to increase the probability of a group reaching binary consensus.
> As one way of measuring whether or not a group has “reached consensus”, we conduct two new analyses which we will include in the appendix and briefly describe in the camera ready version of the main manuscript. First, we examine on each round whether or not a candidate from each of the baseline models achieves unanimous consent, where consent is defined as any degree of agreement (i.e., a participant at least “somewhat agrees” with a candidate consensus).  We look particularly at the divisive questions (those questions for which there was internal disagreement about the Position Statement). Under this analysis, we find that the SFT-Utilitarian model generates a candidate that achieves unanimous consent on 41% of rounds, whereas all other models generate a unanimous consent candidate on 30-33% of rounds.
> Second, we also look at the proportion of candidates that are less divisive than the Position Statements (where candidate divisiveness is the proportion of participants who agree with the statement). It is impossible for a candidate to be less divisive than an Undivisive Position Statement, thus we again only look at the Divisive questions.  We find that 93.5% of candidates generated from the SFT-Utilitarian model are less divisive than the corresponding Position Statements (the SFT-Base model achieves 91.9%, and the prompted models achieve ~85% improvements upon the position statement divisiveness). Together, these results suggest that even in cases that one might expect it to be difficult to find consensus, our model tends to find some common ground.
>
> **“It is also interesting to verify the coverage of generated consensus (i.e., whether it covers all the points that the users agree on)”**
>
> The issue of coverage of the generated consensus is an interesting one. Our task setup is likely too impoverished a setting, however, to elicit text responses that contain points that all users explicitly do or do not agree upon. Consider that users must generate all of their ‘points’ spontaneously in response to our questions (see Table 1 of main text). Our questions touch on rich, multifaceted topics, and participants freely talk about whatever it is that they feel justifies their perspective. For example, in the discussion of speed limits in Table 1, Opinion 1 mentions enforcement and accidents, and Opinion 2 is primarily concerned with pollution and the environment. It is not clear that there are any specific points that these participants agree upon that could be gleaned from the text alone. The virtue and strength of our language modelling with reward modelling approach is that it can try to identify the latent dimensions of commonality between the opinions and generate a statement that the opinion writers would all agree with (see Table 1 consensus as an example).

---

> ### Author Response · Authors · 2022-08-02
> **Response to Reviewer 3uTu (2/2)**
>
> **“The paper uses a quite large language model (70B parameters) for finetuning. Is this necessary and can we use smaller models? Usually, small models can also handle summarization and quality evaluation well if there is enough data, and smaller models are easier to deploy.”**
>
> Thanks very much for this question. We conducted a new experiment to directly address this point, and found that GPT2-sized models can be used for this task (at least when fine-tuned with data from the 70B model), but are worse performing.
>
> We trained a GPT2-sized model (1.4B parameters) on our previously collected data, and conducted a new human evaluation experiment with 245 participants. We directly compared consensus candidates generated by 6 models: i) SFT-Utilitarian 70B, ii) Few-shot 70B, iii) Zero-shot 70B, iv) SFT-Utilitarian 1.4B, v) Few-shot 1.4B, and vi) Zero-shot 1.4B. Here, SFT-Utilitarian corresponds to the fine-tuned model with welfare-based reranking (see paper for details). We chose 1.4B as a smaller baseline model to compare against, as it is GPT-2 sized but it has the same improved architecture and dataset as our 70B model  (see Hoffman et al. 2022). Both the generative model and the reward model are scaled down to 1.4B parameters.
>
> We have included the full figures in a new “Rebuttal.pdf” appendix but we will summarise our findings here. SFT-Utilitarian 70B still significantly outperforms each baseline in win rate when comparing mean agreement. Interestingly, however, SFT-Utilitarian 1.4B performs only slightly worse (the 70B version wins over the 1.4B version 59.4% [53.7%, 64.7%] of the time). This effect size is similar to SFT-Utilitarian 70B versus SFT-Base 70B, suggesting that both scale and reward modelling are independently beneficial for consensus generation. Additionally, SFT-Utilitarian 1.4B outperforms both zero-shot and few-shot 70B (win rate of 75.9% and 75.2% respectively). This is in line with the reviewer’s intuition that smaller models can perform well if there is sufficient data. That is, even though large models can now increasingly be accessed via API or downloaded open source, smaller models are still significantly cheaper and easier to use, and may be able to perform complex tasks like consensus generation when finetuned on sufficiently high quality data. We will add the full 70B to 1.4B comparison (including generated examples and raw Likert scores) to the appendix.
>
> Note, finally, that we did not rerun the full training and data collection pipeline for 1.4B (i.e. we did not first collect human data with 1.4B zero-shot/few-shot and then do two rounds of 1.4B model fine-tuning) but instead fine-tuned with data that we collected using the 70B model. Given that the smaller model has significantly weaker prompting capabilities, running the full pipeline with this small model would have probably made the process much more data intensive, as we could not have bootstrapped the high prompted performance of the 70B model.
>
> J. Hoffmann et al. Training compute-optimal large language models. arXiv preprint arXiv:2203.15556, 2022.
>
> **“There is one step I do not quite understand about model training: in the first round, how is the top-1 candidate chosen from the 16 candidates? In the first round, we do not have the reward model yet, so it seems we do not have the utilities u1,…,un for ranking.”**
>
> We will make this clearer in the paper. On the first iteration, we did not use this sampling scheme. Instead we simply sample a single candidate from a prompted model. We attempted to explain this with the following passage (line 167) which we agree is not sufficiently clear:
> “on the first iteration, we bootstrap first with zero-shot prompting and then with few-shot prompting of the base Chinchilla model.”
> We will replace this with the following:
> “On the first iteration we have no reward model for the reranking and selection scheme. Instead we sample a single candidate from a prompted Chinchilla model (initially zero-shot prompted, then subsequently few-shot prompted - see Appendix).”

---

> ### Comment · Reviewer_3uTu · 2022-08-09
> **Thank you for the reply**
>
> Thank you for the comprehensive reply! I think my concerns are adequately resolved.
>
> Thanks for resolving my question on first-round training! I guess I missed that in my reading.
>
> It is great to see GPT-2 can also work well on this task.

---

### Official Review · Reviewer_5pge · 2022-07-14

**Rating:** 9
**Confidence:** 5
**Soundness:** 4 excellent
**Presentation:** 4 excellent
**Contribution:** 4 excellent

**Summary:**

This paper presents a method of fine-tuning a language model to analyze a set of opinions and produce a candidate consensus statement.  This is accomplished by various human evaluations, re-ranking, and fine-tuning algorithms. The consensus statements produced outperform various baselines, while retaining important information.

**Questions:**

None.

**Limitations:**

The paper does a good job of assessing both limitations and ethical issues.

**Strengths And Weaknesses:**

Wow. I really, really liked this paper. I found it intellectually stimulating, technically complex but understandable, interdisciplinary, and potentially impactful in a way that I rarely see.

+ The paper addresses an important, timely, societally relevant problem
+ The paper makes substantial methodological and technical contributions
+ The paper is well written
+ The results are powerful
+ I believe the paper will serve as a foundational piece of work for subsequent research
+ The paper draws on a wealth of ideas from computer science and social science
+ The paper has carefully considered limitations and ethical concerns
+ The paper has potential for strong, immediate impact if used properly

Weaknesses:

- While I appreciate the technical goal of using reranking to be able to change SWFs at test time, it's unclear if that's actually a useful feature. And while the paper claims that reranking is on par with or exceeds an RL-based approach, it would have been nice to include such a direct, RL based consensus generator as a baseline.
- While the model does outperform all baselines (usually), the difference is not as pronounced as I would like
- The paper does devolve into excessive technicalities at times

---

> ### Author Response · Authors · 2022-08-02
> **Response to Reviewer 5pge (1/1)**
>
> We thank the reviewer for their positive review. We are encouraged by the strengths the reviewer points out and respond to the weaknesses below point-by-point .
>
> **“While I appreciate the technical goal of using reranking to be able to change SWFs at test time, it's unclear if that's actually a useful feature. And while the paper claims that reranking is on par with or exceeds an RL-based approach, it would have been nice to include such a direct, RL based consensus generator as a baseline."**
>
> First, to clarify, we refer in the footnote to two recent papers that compare reranking to RL. These references show that in those settings performance is similar. We do not expect that in our case the performance difference will be very different but we agree that this is an empirical question worth studying and something that we have considered multiple times during this project. We decided against it for two main reasons:
>
> - We aimed to show that language models can be used for an important new task and that, more generally, aligning large language models with the preferences of a group of people is an interesting and worthwhile new avenue to explore. We think that the opinion-conditional reward modelling and the social welfare framework are important contributions in this direction but that reinforcement learning is not a necessary component.
>
> - In addition to not having out-of-the-box flexibility with respect to the SWFs, an RL-based approach would have added considerable technical complexity and would likely have also made our study much more data-intensive. Prior work mitigates the issue of needing large datasets of written text by relying on large external datasets of, for example, articles (if the task is summarization) or questions (if the task is question answering). In our case, where the task is consensus generation, we would have to collect a very large number of questions with corresponding sets of opinions. Given that collecting written data is significantly more time consuming than collecting ratings, collecting a large dataset could make this work prohibitively expensive, at least for a proof-of-principle.
>
> One interesting alternative approach could be to also train an opinion-generating model which generates opinions in response to questions so we could simulate participants and make the pipeline more data efficient. In that case we would need a smaller dataset of opinions and only a large dataset of questions. However, this approach would add a lot of complexity to our method and analysis and is hence outside of the scope of this paper. Nevertheless, this does present a very interesting avenue for future work.
>
> **“While the model does outperform all baselines (usually), the difference is not as pronounced as I would like.”**
>
> This research is based on an entirely new task - that of consensus-generation. As such, we do not have access to a standardised set of baseline measures as is often the case with more established paradigms. All baselines were specifically created with this task in mind, and were intended to act as challenging baselines to beat. However, we have now run an additional baseline evaluation comparing a smaller GPT2-like model (1.4B parameters) against our larger main model (as requested by other reviewers). The prompted versions of this smaller model are weaker baselines, and as expected, our main model (SFT-Utilitarian 70B) drastically outperforms these baselines. We find win rates of 95% [93%, 97%] against the 1.4B zero-shot prompted model and 87% [84%, 91%] versus the 1.4B few-shot prompted model.
>
> **“The paper does devolve into excessive technicalities at times.”**
>
> Thank you for this feedback. Prior to submitting the camera-ready version we will carefully pick through the entire paper and attempt to clarify or remove any egregious technical terms.

---

### Author Response · Authors · 2022-08-08
**Reminder for the end of the Author-Reviewer Discussions**

We wish to thank the reviewers again for helping us improve the paper. We hope that our responses and additional experiments have allayed your concerns. If there is anything that remains outstanding, please let us know as the discussion period ends tomorrow. Thanks again!

---

### Meta-Review · Area_Chair_oxSD · 2022-08-27

**Recommendation:** Accept
**Confidence:** Certain

**Metareview:**

This paper tackles the interesting task of training a language model to generate a consensus statement that maximize the expected  approval of a group of people with diverse opinions. The proposed approach that is based on human evaluations of LM-generated statements and training of a reward model, is compared with a set of baselines. Based on reviewers' suggestions, authors performed additional experiments with smaller models and included analysis of results.

SACs, please note that reviewer bGmJ changed their rating from 3: Reject to 6: Weak Accept verbally, but this change isn't reflected in the average rating.

**Award:**

No

---

### Decision · Program_Chairs · 2022-09-14

Accept